# Modular output circuits of the fastigial nucleus for diverse motor and nonmotor functions of the cerebellar vermis

**Hirofumi Fujita[1]\*, Takashi Kodama[1], Sascha du Lac[1,2,3]\***

[1]Department of Otolaryngology-Head and Neck Surgery, Johns Hopkins University, Baltimore, United States; [2]Department of Neuroscience, Johns Hopkins University, Baltimore, United States; [3]Department of Neurology, Johns Hopkins Medical Institute, Baltimore, United States

**Abstract** The cerebellar vermis, long associated with axial motor control, has been implicated in a surprising range of neuropsychiatric disorders and cognitive and affective functions. Remarkably little is known, however, about the specific cell types and neural circuits responsible for these diverse functions. Here, using single-cell gene expression profiling and anatomical circuit analyses of vermis output neurons in the mouse fastigial (medial cerebellar) nucleus, we identify five major classes of glutamatergic projection neurons distinguished by gene expression, morphology, distribution, and input-output connectivity. Each fastigial cell type is connected with a specific set of Purkinje cells and inferior olive neurons and in turn innervates a distinct collection of downstream targets. Transsynaptic tracing indicates extensive disynaptic links with cognitive, affective, and motor forebrain circuits. These results indicate that diverse cerebellar vermis functions could be mediated by modular synaptic connections of distinct fastigial cell types with posturomotor, oromotor, positional-autonomic, orienting, and vigilance circuits.

**\*For correspondence:**
hirofumi@jhmi.edu (HF);
sascha@jhmi.edu (SdL)

**Competing interests:** The authors declare that no competing interests exist.

## Introduction

The cerebellum has been implicated in several cognitive functions and neuropsychiatric disorders, which are more typically associated with the cerebral cortex and basal ganglia (*Tsai et al., 2012*; *Hariri, 2019*; *Schmahmann et al., 2019*). Although it is widely assumed that such nonmotor functions are mediated by the cerebellar hemispheres and their connections with the thalamus (*Strick et al., 2009*; *Buckner, 2013*; *Wang et al., 2014*; *De Schutter, 2019*), increasing functional and anatomical evidence points to roles for the vermis (*Schmahmann and Sherman, 1998*; *Watson et al., 2009*; *Halko et al., 2014*; *Watson et al., 2014*; *Zhang et al., 2016*; *Wagner et al., 2017*; *Badura et al., 2018*; *Xiao et al., 2018*; *Albazron et al., 2019*; *Brady et al., 2019*; *Watson et al., 2019*; *Kelly et al., 2020*) an evolutionarily old portion of the cerebellum best known for its influence on brainstem circuits for posture and eye movements (*Chambers and Sprague, 1955*; *Ohtsuka and Noda, 1991*). Structural and functional abnormalities of the vermis have been associated with various psychiatric disorders and conditions, including autism, schizophrenia, mood disorders, chronic pain, and addiction (*Weinberger et al., 1980*; *Courchesne et al., 1988*; *Sweeney et al., 1998*; *Strakowski et al., 2005*; *Andreasen and Pierson, 2008*; *Moulton et al., 2010*; *Fatemi et al., 2012*; *Moulton et al., 2014*). Remarkably little is known, however, about the specific cell types and circuits responsible for diverse nonmotor functions of the vermis.

The cerebellum is thought to be organized in parallel circuit modules that comprise specific sets of inferior olive (IO) neurons, Purkinje cells (PCs), and vestibular or cerebellar nucleus neurons (*Oscarsson, 1979*; *Ito, 1984*; *Apps and Hawkes, 2009*; *Voogd, 2011*; *Apps et al., 2018*). The

cerebellar vermis, caudal medial accessory olive, and fastigial nucleus (FN; medial cerebellar nucleus) collectively constitute the broad 'A' module (*Groenewegen and Voogd, 1977*), which is likely to include several submodules (*Apps, 1990*; *Sugihara and Shinoda, 2004*; *Voogd and Ruigrok, 2004*; *Sugihara and Shinoda, 2007*). Although the fastigial nucleus has been implicated in diverse motor and nonmotor functions in human imaging studies (*Schmahmann, 1991*; *Schmahmann, 1998*; *Albazron et al., 2019*), its small size (*Diedrichsen et al., 2011*; *Tellmann et al., 2015*) has precluded precise functional localization. Mono- and multi-synaptic connections with the midbrain, hippocampus, basal ganglia, and cerebral cortex are likely to mediate vermis and fastigial influences on affective and cognitive functions (*Steriade, 1995*; *Teune et al., 2000*; *Watson et al., 2014*; *Badura et al., 2018*; *Xiao et al., 2018*; *Watson et al., 2019*; *Vaaga et al., 2020*) but the specific fastigial cell types responsible for these influences are not known.

In this study, we hypothesized that the diverse motor and nonmotor functions of the cerebellar vermis are mediated by multiple types of fastigial output neurons with distinct circuit connections. By combining single-cell gene expression, immunohistochemical, and circuit connectivity analyses, we identified several molecularly and anatomically distinct types of excitatory projection neurons in the mouse fastigial nucleus. Anterograde, transsynaptic, and retrograde tracing analyses indicate segregated, modular circuit connectivity; each fastigial cell type is linked with a specific set of PCs and IO neurons and makes divergent projections to functionally-related downstream targets. Disynaptic connections with the basal ganglia, basal forebrain, and cerebral cortex revealed that specific fastigial cell types are linked with circuits responsible for distinct aspects of cerebellar motor and nonmotor functions.

## Results

### Anatomically distinct cell types in the fastigial nucleus

The fastigial nucleus (FN) is the medial cluster of cerebellar nucleus neurons (*Dow, 1942*), delineated in mammals by surrounding white matter (*Larsell, 1970*). Despite its small size, the FN comprises at least three subdivisions (*Korneliussen, 1968*; *Beitz and Chan-Palay, 1979*): a rostral part, a caudal part, and a dorsolateral protuberance (DLP), which is well developed in rodents (*Fujita et al., 2010*). Each of these subdivisions have been linked with distinct sets of Purkinje cells, inferior olive subnuclei, and downstream brain regions (*Batton et al., 1977*; *Teune et al., 2000*; *Voogd and Ruigrok, 2004*; *Sugihara and Shinoda, 2007*; *Sugihara et al., 2009*).

To identify markers that could distinguish cell types in the FN, we searched in situ hybridization data in the Allen Brain Atlas (http://mouse.brain-map.org/) (*Lein et al., 2007*) for genes expressed differentially across FN subregions. We then evaluated candidate marker genes (*Figure 1—figure supplement 1*) with immunostaining and confocal microscopic analyses of serial cerebellar sections. Antibodies for SPP1 (osteopontin), SNCA (alpha-synuclein), and CALB2 (calretinin) reliably labeled specific subsets of neurons that were located predominantly in distinct subregions of the FN (*Figure 1*; *Figure 1—figure supplement 2*). In caudal regions of the FN (*Figure 1B*), SPP1-immunopositive (SPP1+) neurons were prominent in the caudal portion of the DLP (cDLP) and were sparsely distributed ventrally. In contrast SNCA+ neurons were prominent ventrally in the caudal fastigial nucleus (cFN), but were sparse in the cDLP. In rostral regions of the FN (*Figure 1C*), the majority of neurons were SPP1+. Neurons strongly immunopositive for CALB2 (CALB2+) were prominent in rostrally in ventral and lateral parts of the fastigial nucleus (vlFN).

Double-immunostaining revealed five populations of neurons that were distinguished by their marker gene expression and anatomical distribution. Neurons that exclusively expressed SPP1 were found in two distinct regions: the rostral FN (rFN) (*Figure 1E*) and the rostral DLP (rDLP). In contrast, neurons that coexpressed SPP1 and SNCA (*Figure 1H*) were found in the cDLP (*Figure 1—figure supplement 2*). They were also scattered through the cFN, where they intermingled with a distinct population of neurons which exclusively expressed SNCA (*Figure 1F*). CALB2+ neurons (*Figure 1G*), which were distributed in the vlFN, coexpressed SNCA but not SPP1.

Although the majority of neurons in the cerebellar nuclei are glutamatergic, distinct populations of glycinergic and GABAergic neurons have been reported (*Fredette and Mugnaini, 1991*; *Bagnall et al., 2009*). To identify neurotransmitter associated with each cell type, we performed immunostaining for SNCA, SPP1, and CALB2 on cerebellar nucleus sections from mouse lines that

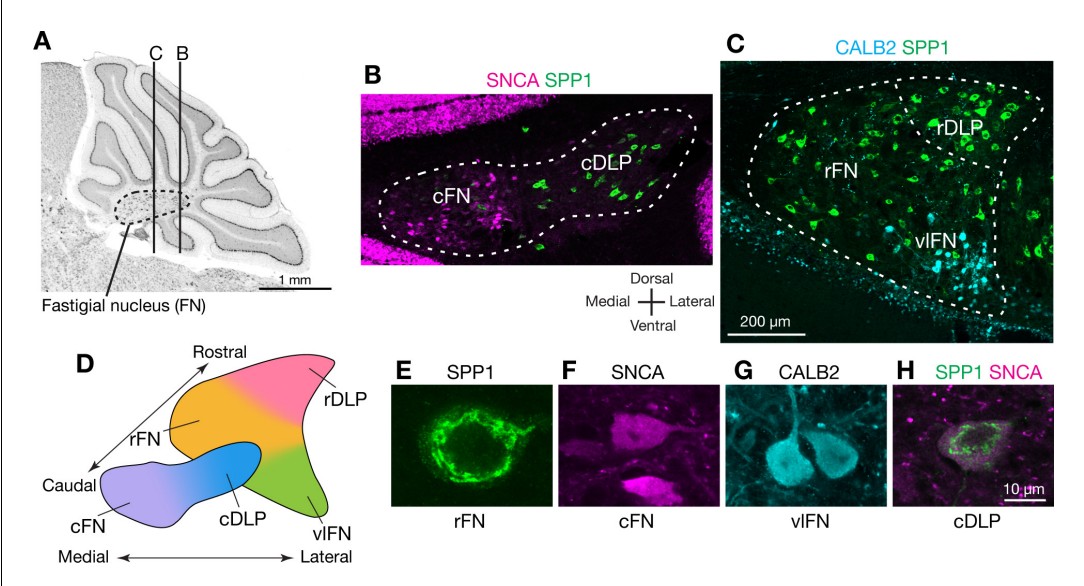

**Figure 1.** Anatomically distinct cell types in the fastigial nucleus. (**A**) The fastigial nucleus (FN) in a sagittal section of the mouse cerebellar vermis identified with Nissl staining. Vertical lines indicate the location of sections in panels B and C. (**B**) Double immunostaining for alpha synuclein (SNCA) and osteopontin (SPP1) in a coronal section of the FN (n = 3 males). SNCA immunopositive neurons (green) are prominent in the caudal FN (cFN). SPP1 immunopositive neurons (magenta) are distributed throughout the caudal portion of the dorsolateral protuberance (cDLP). (**C**) Double immunostaining for calretinin (CALB2) and SPP1 in a more rostral section of the FN (n = 3 males). Intensely CALB2+ neurons (cyan) are densely clustered in the ventrolateral FN (vlFN). SPP1+ neurons (red) are distributed throughout the rostral FN (rFN) and the rostral portion of the DLP (rDLP). (**D**) Five subregions of the FN can be delineated by the distribution of neurons expressing SNCA, CALB2, and/or SPP1: (1) the cFN (purple) comprises SNCA+ neurons; (2) the cDLP (blue) comprises neurons which co-express SNCA and SPP1; (3) the vlFN (green) comprises neurons that express CALB2 and SNCA; (4, 5) the rDLP (pink) and rFN (yellow) each comprise exclusively SPP1+ neurons. (**E–H**) High magnification images of immunohistochemically revealed neurons located in the rFN (E, SPP1), cFN (F, SNCA), vlFN (G, CALB2) and cDLP (H, SPP1 and SNCA). Note considerable difference in the sizes of these neurons. Scale bar in C applies to B and C. Scale bar in H applies to E-H.

The online version of this article includes the following figure supplement(s) for figure 1:

**Figure supplement 1.** Expression of marker candidate genes for fastigial cell types.

**Figure supplement 2.** Distribution of molecularly distinct fastigial neurons that were identified with immunostaining for SPP1, CALB, and SNCA.

**Figure supplement 3.** Overlap of fastigial cell type markers with glutamatergic, glycinergic, and GABAergic neurons.

express fluorescent reporters for glutamatergic neurons (VgluT2-Cre;Ai14), GABAergic neurons (Gad2-nls-mCherry) and glycinergic neurons (GlyT2-EGFP) (*Figure 1—figure supplement 3*). Notably, most (93%; n = 954) of the identified glutamatergic neurons, but none of the GABAergic neurons, were immunostained by either SPP1 or SNCA (*Figure 1—figure supplement 3A–C,E*). Glycinergic neurons in the FN comprise two distinct populations *Bagnall et al., 2009*; we found that large glycinergic projection neurons, located ventrally in the rFN, were robustly immunostained by SPP1 (but not SNCA) (*Figure 1—figure supplement 3D*). In contrast, small glycinergic neurons were not labeled by either SPP1 or SNCA (*Figure 1—figure supplement 3D*). Collectively, these results indicate that the majority of glutamatergic fastigial neurons can be classified into five major types by localization and expression of SPP1, SNCA, and CALB2.

## Single cell gene expression confirms anatomically distinct cell types

To verify the anatomical analyses of cell types, we performed single-cell qPCR analyses on acutely isolated fastigial cells using a strategy previously developed for distinguishing cell classes among vestibular nucleus neurons (*Figure 2—source data 1*; *Kodama et al., 2012*). From an initial pool of 130 randomly harvested fastigial cells, we identified 50 cells which expressed ion channel genes associated with action potentials (*Scn8a* and *Kcna1)* and lacked non-neuronal markers (*Mobp* and *Cd68*), and were thus considered as neurons. Most of these neurons (84%: 42/50) expressed *Slc17a6* (vesicular glutamate transporter 2, VgluT2, *Figure 2A*), indicating that they were glutamatergic. The

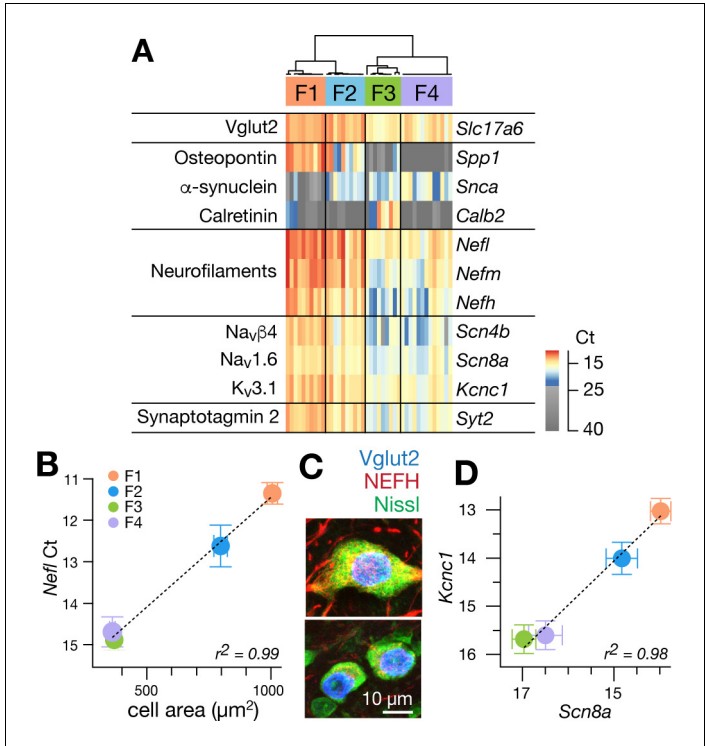

**Figure 2.** Single cell gene expression analyses confirm molecularly distinct cell types. (**A**) Heatmap representation of quantitative gene expression profiles obtained via single-cell qPCR for individual excitatory fastigial neurons that express *Slc17a6* (=*VgluT2*). Expression levels in Ct (cycle threshold in qPCR) are color coded, where insignificant expression (<5 copies of transcript, corresponding to Ct of 23.45) is shown in grey scale. Columns and rows correspond to individual neurons (n = 42) and genes examined, respectively. Clustering analysis for expression of *Spp1*, *Snca*, and *Calb2* confirms four major types of fastigial neurons immunohistochemically identified, which are termed F1-F4 as shown in the dendrogram. Neurons included are from 7 (6 wildtype and 1 YFP-16) male mice. (**B**) Positive correlation of *Nefl* expression in the molecularly defined cell types with cell body area measured from the corresponding neurons immunohistochemically identified (n = 210). Plots are color-coded for the cell-types as indicated in A (F1, orange; F2, blue; F3, green; F4, purple). Note that smaller Ct values indicate greater expression levels. Population averaged data for each cell type are plotted. Error bars represent SEM. (**C**) NEFH immunostaining of glutamatergic fastigial neurons identified by nuclear-localized GFP (blue) in VgluT2-Cre;SUN1-sfGFP line. Size of the somata is identified with Nissl staining (green). Immunoreactivity for NEFH (red) is higher in the large glutamatergic neurons (top) than in the small glutamatergic neurons (bottom). (**D**) Linear correlation between expression levels of *Scn8a* vs *Kcnc1*. Color-code of the cell-type is the same as B. Plotted are population averaged data for each cell type. Error bars represent SEM.

The online version of this article includes the following source data and figure supplement(s) for figure 2:

**Source data 1.** Raw data of single-cell qPCR with excitatory fastigial neurons on selected genes Gene expression levels (in qPCR Ct) in the individual neurons are organized in columns.

**Figure supplement 1.** Correlation of neurofilament gene expression with axonal caliber and the expression of ion channels and synaptic molecules.

---

remainder expressed inhibitory neuronal markers *Gad1*, *Gad2*, and/or *Slc6a5*. We focused subsequent expression analyses on the glutamatergic excitatory neurons.

Nearly all excitatory neurons (97.6%: 41/42) significantly expressed at least one of the cell-type markers (*Spp1*, *Snca*, and *Calb2*), as confirmed with immunostaining in VgluT2-Cre;Ai14 line (*Figure 1—figure supplement 3C*). Hierarchical clustering based on these markers revealed four cell types in the fastigial glutamatergic population, which we termed F1-F4 (*Figure 2A*). Two cell types expressed *Spp1* and were distinguished by the absence (F1) or presence (F2) of *Snca*. The other two cell types both expressed *Snca* and were separated by strong *Calb2* expression in F3 but not F4. The combinatorial marker expression patterns corresponded with those observed in immunostaining

(*Figure 1*): F1 with Spp1+ neurons in the rFN (F1$_R$) and the rDLP (F1$_{rDLP}$), F2 with Spp1+/Snca+ neurons in the cDLP, F3 with Calb2+/Snca+ neurons in the vlFN, and F4 with Snca+ neurons in the cFN.

To further assess distinctions across cell types, we examined expression levels of neurofilament genes. *Nefl*, *Nefm*, and *Nefh* were highest in F1, intermediate in F2, and lowest in F3 and F4 neurons (*Figure 2A*), suggesting that the axon diameter of these cell types are in the order of F1 >F2>F3/F4 (*Friede and Samorajski, 1970*; *Hoffman et al., 1987*; *Lee and Cleveland, 1996*). Differences across cell types in axonal caliber were confirmed with local injections of adeno-associated virus (AAV.hSyn. TurboRFP) into the rFN (predominantly F1) and cFN (predominantly F4) (*Figure 2—figure supplement 1A–C*). As predicted, axons from the rFN were thicker than those from the cFN (1.9 ± 0.6 s.d. μm vs 1.0 ± 0.3 s.d. μm, n = 30 vs. 25, respectively). Neurofilament expression levels were also well correlated with cell body area assessed in immunohistochemically identified fastigial cell types (*Nefl* in *Figure 2B*; $r^2$ = 0.996, p=0.002; *Nefm* and *Nefh* in *Figure 2—figure supplement 1D–F*).

Excitatory projection neurons in the cerebellar nuclei can be distinguished from local interneurons and inhibitory nucleo-olivary neurons by their ability to fire at high rates (*Uusisaari et al., 2007*; *Bagnall et al., 2009*; *Najac and Raman, 2015*). Each of the excitatory FN neurons expressed *Scn8a* (Nav1.6), *Scn4b* (Navb4), *Kcnc1* (Kv3.1), and *Syt2* (synaptotagmin 2), as predicted for fast-firing neurons (*Lien and Jonas, 2003*; *Mercer et al., 2007*; *Kodama et al., 2012*; *Kodama et al., 2020*). Previous studies in vestibular nucleus neurons, which share multiple properties with cerebellar nucleus neurons, including high spontaneous firing rates (*Uusisaari et al., 2007*; *Bagnall et al., 2009*; *Najac and Raman, 2015*) and direct innervation by Purkinje cells (*Sekirnjak et al., 2003*; *Shin et al., 2011*), demonstrated that variations in neuronal firing capacities can be predicted by the absolute expression levels of these and other 'fast-spiking' genes (*Kodama et al., 2020*). Fastigial cell types F1-F4 exhibited different expression levels of action potential genes *Scn8a* and *Kcnc1*, but with an almost constant ratio ($r^2$ = 0.98, *Figure 2D*). *Nefl*, *Nefm*, *Nefh*, *Scn4b*, and *Syt2* followed the same trend (*Figure 2—figure supplement 1G–I*); absolute expression levels of these genes were consistently in the order of F1 >F2>F3/F4, suggesting that F1, F2, and F3 and F4 would have the fastest, intermediate, and the slowest firing capacities.

## Downstream targets of excitatory fastigial cell types are distinct

Although projections from rostral vs caudal fastigial nucleus are known to differ (*Angaut and Bowsher, 1970*; *Bentivoglio and Kuypers, 1982*), little is known about the downstream targets of specific fastigial cell types. We performed a series of anterograde, transsynaptic, and retrograde anatomical tracing experiments to identify global and cell-type specific connectivity patterns. Pan-fastigial injections of anterograde tracer AAV9.hSyn.TurboRFP (*Figure 3A*) revealed divergent fastigial projections to over 60 distinct brain regions (*Figure 3* and *Figure 3—figure supplement 1*, *Figure 3—source data 1*), which were confirmed with complementary injections of anterograde transsynaptic tracer AAV1.hSyn.Cre in Ai14 reporter mice (*Zingg et al., 2017*; *Figure 3A*). Projection patterns identified in these pan-fastigial injections were quite consistent across individuals (n = 6 males for axonal labeling; n = 4 male and n = 3 female for transsynaptic labeling). Fastigial axonal terminals and postsynaptic neurons contralateral to injection sites were prominent in cervical spinal cord (*Figure 3C*), cerebellar cortex (*Figure 3—figure supplement 1F*), and several regions of the medulla (*Figure 3D*), pons (*Figure 3—figure supplement 1C*), midbrain (*Figure 3E*, *Figure 3—figure supplement 1D*), and diencephalon (*Figure 3F,G*, *Figure 3—figure supplement 1E*). Notably, fastigial projections to the thalamus were not limited to the 'motor thalamus' (ventrolateral (VL) and ventromedial (VM) nuclei), but also robustly to intralaminar thalamus (centrolateral (CL) and parafascicular (PF)) and the mediodorsal (MD) nucleus (*Figure 3F,G* and *Figure 3—figure supplement 1E*). These projections were considered to derive from excitatory neurons because inhibitory neurons target different brain regions; glycinergic FN neurons project ipsilaterally, to hindbrain and spinal cord (*Bagnall et al., 2009*), and GABAergic FN projections exclusively target the IO, as confirmed with selective anterograde tracing by injecting Cre-dependent AAV into the FN of Gad2Cre mice (*Figure 3—source data 1*).

To identify linkages between specific fastigial cell types and downstream target nuclei, we performed localized anterograde tracing experiments via stereotaxic injections of AAVs (AAV9.hSyn. eGFP, AAV9.hSyn.TurboRFP, and AAV1.hSyn.Cre) into anatomically defined subregions of the FN (*Figure 1*; *Figure 3—source data 2*). Subsequent retrograde tracing combined with immunostaining for the fastigial cell-type markers was performed to confirm FN cell types of origin.

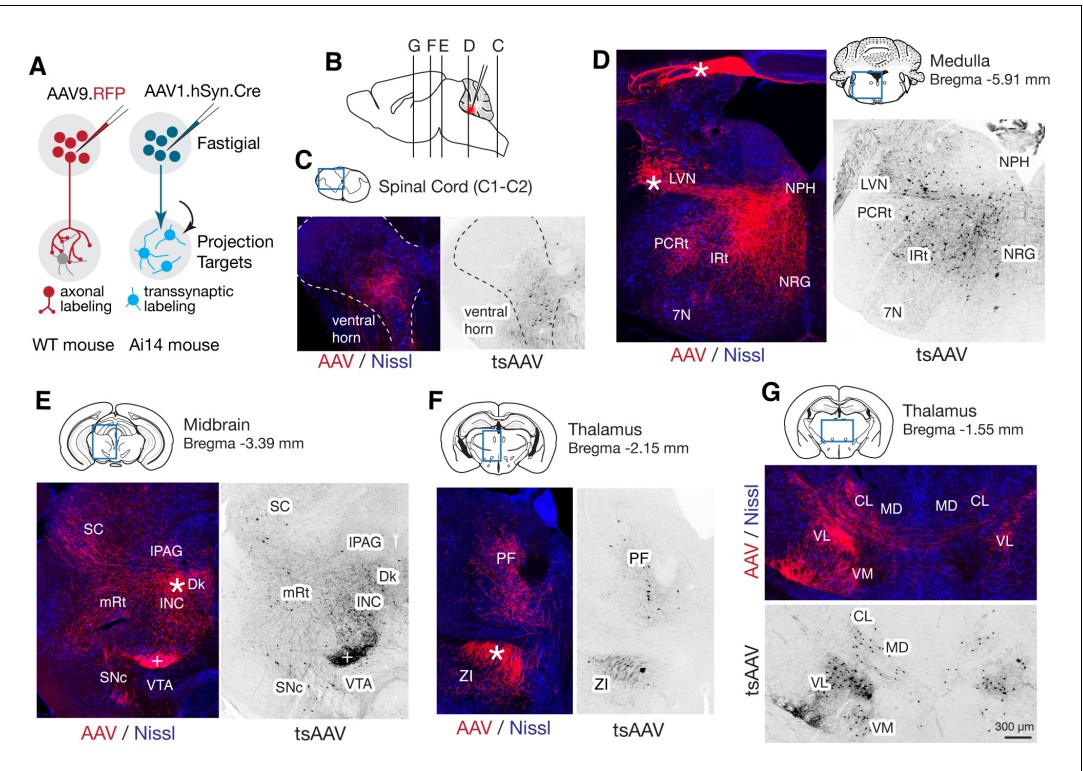

**Figure 3.** Pan-fastigial AAV injections reveal widespread fastigial output. (**A**) Schematics of AAV injection experiments to label fastigial axons and terminals (left) and to transsynaptically label postsynaptic neuronal somata (right). (**B**) Schematic of a sagittal view of the mouse brain to show approximate levels of the coronal sections in C-G. Tracer injection to the FN is illustrated in red. (**C–G**) Representative low magnification confocal images obtained from pan-fastigial injections of AAVs (AAV, AAV9.RFP; tsAAV, AAV1.hSyn.Cre). Contralateral side to the injection site is shown. Blue rectangles in the drawings of the sections indicate imaged areas. In these paired images, the left image shows labeled fastigial axons and terminals in red and counterstained with Nissl in blue; the right image shows transsynaptically labeled neurons in black. Asterisks indicate axonal bundles. Results from these axonal labeling and transsynaptic tracing experiments were consistent across mice (n = 6 males for axonal labeling; n = 4 male and n = 3 female for transsynaptic labeling. (**C**) Shows fastigial projections to the cervical spinal cord, dense at the lamina VII-VIII, at the level of C1-C2. Dotted lines circumscribe the gray matter. (**D**) Shows fastigial projections to the medulla. Labeled fastigial axons crossed the midline within the cerebellum, exit via the cerebellar peduncle and provide dense projections to the vestibular and reticular nuclei, including the lateral vestibular nucleus (LVN), nucleus reticularis gigantocellularis (NRG), intermediate reticular nucleus (IRt), and parvocellular reticular nucleus (PCRt), with additional collateralization to the nucleus prepositus hypoglossi (NPH) and facial nucleus (7N). (**E**) Shows fastigial projections to the midbrain. Labeled fastigial axons innervate perioculomotor nuclei including the interstitial nucleus of Cajal (INC), lateral periaqueductal gray (lPAG), and nucleus of Darkschewitsch (Dk), and more laterally located nuclei including the superior colliculus (SC), mesencephalic reticular nucleus (mRt), and substantia nigra pars compacta (SNc). Innervation of the ventral tegmental area (VTA) is very sparse. Note that the dense labeling in the red nucleus (+) derived from axons from the anterior interpositus that were labeled by AAV that leaked from the injection center in the FN (*Figure 3—figure supplement 1B*, bottom). Injections specifically localized to FN subregions did not significantly label the red nucleus (*Figure 5—figure supplement 1*). (**F and G**) Fastigial projections to the thalamus. Labeled fastigial terminals and transsynaptically labeled somata are distributed at the PF, CL, MD, VM, and VL thalamic nuclei and the zona incerta (ZI). Fastigial axons projecting to the contralateral thalamus traverse the midline at the level of G, and innervate the ipsilateral thalamus. Scale bar in G applies to all confocal images in C-G. Abbreviations, 7N, facial nucleus; CL, centrolateral thalamic nucleus; Dk, nucleus of Darkschewitsch; INC, interstitial nucleus of Cajal; IRt, intermediate reticular nucleus; lPAG, lateral periaqueductal gray; LVN, lateral vestibular nucleus; MD, mediodorsal thalamic nucleus; mRt, mesencephalic reticular nucleus; NRG, nucleus reticularis gigantocellularis; NPH, nucleus prepositus hypoglossi; PCRt, parvocellular reticular nucleus; PF, parafascicular thalamic nucleus; SC, superior colliculus; SNc, substantia nigra pars compacta; VL, ventrolateral thalamic nucleus; VM, ventromedial thalamic nucleus; VTA, ventral tegmental area; ZI, zona incerta.

The online version of this article includes the following source data and figure supplement(s) for figure 3:

**Source data 1.** Fastigial projection targets identified by localized anterograde tracer injections.

**Source data 2.** List of tracers, coordinates, injection volume, and mice used for tracing experiments.

**Figure supplement 1.** Fastigial projection targets revealed with AAV- mediated anterograde tracing.

**Figure supplement 2.** Cre-dependent anterograde tracing of GABAergic neuronal projections from the fastigial nucleus.

## Rostral fastigial projection targets

To distinguish projections from $F1_R$, F3, and $F1_{rDLP}$, we made localized injections of AAVs into the rFN, vlFN, and rDLP, respectively, which resulted in differential terminal labeling in several regions of the hindbrain, including nucleus reticularis gigantocellularis (NRG), ventral medullary (MdV) and intermediate (IRt) portions of the reticular formation, and inferior vestibular nucleus (IVN) (*Figure 4A,B*, *Figure 3—source data 1*). Distinct projections from each cell type were confirmed with complementary retrograde tracing experiments in which FastBlue or retrobeads were injected into NRG/MdV, IVN, and IRt, which are the major projection targets of $F1_R$, F3, and $F1_{rDLP}$, respectively. Those injections resulted in labeling of SPP1+ large neurons in the rFN (*Figure 4C*), CALB2+ small neurons in the vlFN (*Figure 4D*), and SPP1+ large neurons in the rDLP (*Figure 4E*).

Anterograde axonal tracing paired with transsynaptic tracing identified the projections of $F1_R$, F3, and $F1_{rDLP}$ as follows. Consistent with the known functions of the rostral fastigial nucleus in posture and locomotion (*Chambers and Sprague, 1955*; *Mori et al., 1999*), $F1_R$ neurons in the rFN made extensive projections to the spinal cord, MdV, NRG, lateral paragigantocellularis (LPGi) (*Capelli et al., 2017*), IVN, and lateral vestibular nucleus (LVN) (*Figure 4—figure supplement 1*), as summarized in *Figure 4J* and *Figure 3—source data 1*. Axonal terminals of $F1_R$ neurons densely innervated the somata and proximal dendrites of huge neurons (diameter >35 μm) in the LVN and NRG (*Figure 4F*), providing a circuit substrate for rapid cerebellar modulation of vestibulospinal, reticulospinal, and reticuloarousal neurons (*Eccles et al., 1975a*; *Eccles et al., 1975b*; *Wilson and Peterson, 1978*). Brainstem autonomic nuclei implicated in cardiovascular and respiratory functions, including the paramedian nucleus (PMn, *Figure 4F*; *Figure 4—figure supplement 1*) and LPGi, were also targeted by $F1_R$ neurons, as expected from physiological studies in cats (*Reis et al., 1973*; *Bradley et al., 1987*). A previously undescribed projection from the rFN to the subcoeruleus region of the dorsal pons (*Figure 4F*) was revealed by transsynaptic labeling in tyrosine hydroxylase (TH) immunopositive neurons, which are known to comprise the coeruleospinal pathway (*Fung et al., 1991*; *Jones, 1991*).

In contrast with the extensive postsynaptic targets of $F1_R$ neurons, the projections of F3 neurons in the vlFN were restricted to the dorsal aspect of the medulla (*Figure 4—figure supplement 1*), as summarized in *Figure 4K* and *Figure 3—source data 1*. Labeled terminals were identified in the Kölliker-Fuse nucleus (KF; *Figure 4G*), which has been implicated in postural and locomotory modulation of respiration (*Giraudin et al., 2012*), and in the medial portion of the IVN (*Figure 4A*), which is associated with autonomic function (*McCall et al., 2017*). Of note, although both $F1_R$ and F3 neurons project to the IVN, their postsynaptic targets differ in size and location, with large lateral vs small medial IVN neurons targeted by $F1_R$ and F3 neurons, respectively (*Figure 4H*; somata area, $299 \pm 117$ s.d. $\mu m^2$ vs $208 \pm 121$ s.d. $\mu m^2$, n = 40 and 44 from two injections each; t(82) = 3.06, p=0.001, Welch's t-test).

$F1_{rDLP}$ neurons in the rDLP projected to the brainstem regions associated with orofacial motor control, including lateral medullary regions (IRt and PCRt; *Figure 4B*; *Figure 4—figure supplement 1*) and the facial motor nucleus, as summarized in *Figure 4L* and *Figure 3—source data 1*. Retrograde labeling from injections in the IRt labeled exclusively SPP1+ neurons in the rostral DLP ($F1_{rDLP}$, *Figure 4E*). Transsynaptic labeling demonstrated that $F1_{rDLP}$ neurons synapse onto CHAT+ neurons in the facial nucleus (*Figure 4I*), congruent with a previous report demonstrating monosynaptic inputs from the FN to facial motoneurons (*Stanek et al., 2014*).

## Caudal fastigial neuronal targets

Anterograde tracing localized to caudal regions of the FN indicated projections to the brainstem, midbrain, and thalamus (*Figure 5*; *Figure 5—figure supplement 1*). To distinguish projections from caudally located fastigial cell types, we made paired AAV injections targeted to the cDLP (exclusively F2) and the cFN (predominantly F4, with scattered F2; *Figure 1B*). These injections revealed predominantly non-overlapping fastigial thalamic projections (*Figure 5A*); VM, CL, and MD nuclei were innervated by axons originating from the cFN whereas the VL nucleus was innervated by axons from the cDLP. To determine whether these segregated projections originated from distinct fastigial cell types, we made retrobead injections to the VM, CL, or VL thalamus and performed immunostaining for fastigial cell type markers. Retrogradely labeled neurons from CL and VM injections were found in the cFN and expressed SNCA (*Figure 5B*), confirming their identity as F4. In contrast, VL

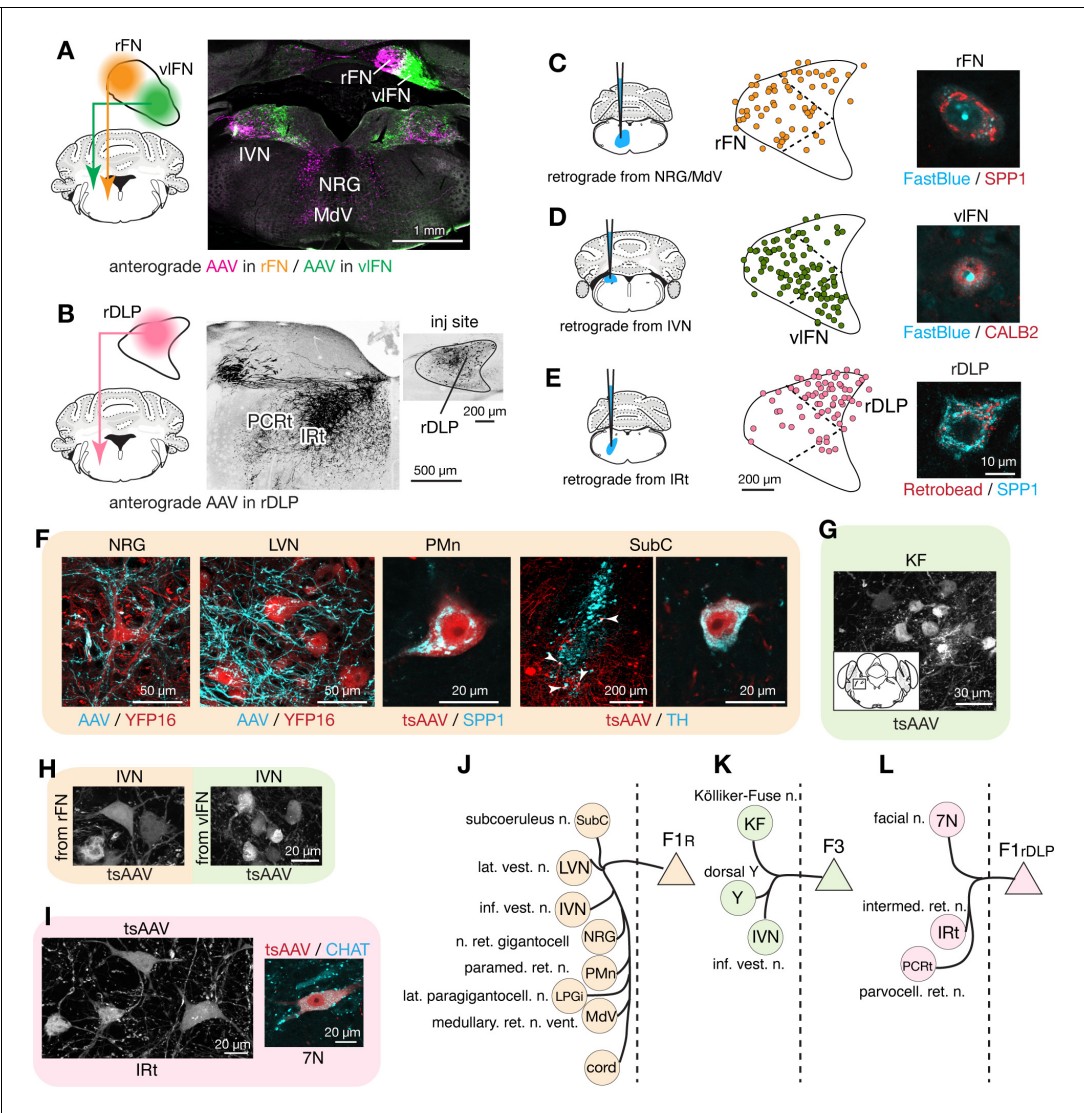

**Figure 4.** Segregated output channels from rostral parts of the FN. (**A and B**) AAV-mediated anterograde tracing with injections localized to subregions of the rostral fastigial nucleus that distinguish projections from $F1_R$, $F3$, and $F1_{rDLP}$ neurons. (**A**) Shows a dual AAV injection in the rFN ($F1_R$ region, AAV9.RFP, magenta in confocal image) and vlFN ($F3$ region, AAV9.GFP, green in confocal image) and the resultant labeling in the inferior vestibular nucleus (IVN) and medulla. Results were consistent across four mice (n = 2 male and n = 2 female) for rFN projections, and across three male mice for vlFN projections. (**B**) Shows an AAV9.RFP injection in the rDLP ($F1_{rDLP}$ region, inset) and the resultant labeling (black in confocal image) in the IRt and PCRt. Results were consistent across three mice (n = 2 male and n = 1 female). (**C–E**) Retrograde tracing experiments to confirm segregation among rostral fastigial output projections. Injections were made into NRG/MdV (C: n = 3 males), IVN (D: n = 3 males), and IRt (D: n = 2 males). (**C**) Shows retrograde tracer (FastBlue, cyan in the right image) injection to the NRG/MdV. Retrogradely labeled neurons are mapped onto the rFN (middle) and overlapped with SPP1+ (red in the right image) neurons of large cell bodies, indicating that $F1_R$ neurons project to the NRG/MdV. Similarly, (**D and E**) Show results of retrograde labeling from the IVN (FastBlue) and IRt (retrobead). The localization of labeled cells and immunoreactivity for CALB2 and SPP1 indicate that $F3$ and $F1_{rDLP}$ neurons project to the IVN and IRt, respectively. Dotted lines in the middle panels indicate approximate borders for the fastigial subregions. (**F**) Key projection targets of $F1_R$ neurons. AAV and tsAAV indicate labeling by AAV9.RFP and AAV1.hSyn.Cre, respectively. Panels show dense fastigial axonal innervation (cyan) of large neurons in the NRG and LVN, which are fluorescently labeled in YFP16 mouse line (red). Also shown are transsynaptically labeled SPP1+ PMn neuron and a TH+ SubC neuron (white arrowheads) (axonal tracing, n = 2 males and n = 2 females; transsynaptic tracing, n = 3 females). (**G**) $F3$ neurons project to the Kölliker-Fuse (KF) nucleus, as demonstrated by anterograde transsynaptic labeling from the vlFN. Inset shows location of the labeled KF neurons (axonal tracing, n = 2 males; transsynaptic tracing, n = 2 males). (**H**) $F1_R$ and $F3$ differentially innervate large and small neurons in the IVN, respectively, as demonstrated by anterograde transsynaptic labeling from the rFN and vlFN. (**I**) Projections of $F1_{rDLP}$ neurons to IRt and CHAT+ facial nucleus neurons, demonstrated by anterograde transsynaptic tracing from the rDLP (axonal tracing, n = 2 males and n = 1 female; transsynaptic tracing, n = 3 females). (**J–L**) Summary of the major output targets of $F1_R$, $F3$, and $F1_{rDLP}$, respectively. Dotted vertical line indicates midline. Targets are arranged rostro-caudally from top to bottom. Scale bars in the middle and right panels in E applies to similar panels in C and D. Scale bar in H applies to both left and right images in H. *Figure 3—source data 1* contains a complete list of

*Figure 4 continued on next page*

Figure 4 continued

projection targets. Abbreviations, 7N, facial nucleus; AAV, adeno associated virus; cord, spinal cord; IRt, intermediate reticular nucleus; IVN, inferior vestibular nucleus; KF, Kölliker-Fuse nucleus; LPGi, lateral paragigantocellular nucleus; LVN, lateral vestibular nucleus; MdV, medullary reticular nucleus, ventral; NRG, nucleus reticularis gigantocellularis; PCRt, parvocellular reticular nucleus; PMn, paramedian reticular nucleus; SubC, subcoeruleus nucleus; tsAAV, anterograde transsynaptic labeling with AAV; Y, dorsal group Y.

The online version of this article includes the following figure supplement(s) for figure 4:

**Figure supplement 1.** Representative results of anterograde tracing from injections localized to rFN, vlFN, or rDLP.

injections retrogradely labeled neurons in the cDLP that expressed SPP1 (*Figure 5C*), confirming their identity as F2.

Localized tracing experiments indicated that F2 neurons in the cDLP projected to midbrain and pontine nuclei associated with orienting movements of the eyes and head, including the superior colliculus (SC), interstitial nucleus of Cajal (INC), caudal pontine reticular formation (PnC), periaqueductal gray (PAG), brainstem nucleus reticularis gigantocellularis (NRG), and cervical spinal cord (*Figure 5D* and *Figure 3—source data 1*). Retrograde labeling confirmed that projections to SC, NRG, and PAG originated from F2 neurons (SPP1+/medium sized cFN neurons). These projections were strikingly consistent with those of the fastigial oculomotor region in primates (*Noda et al., 1990*). F2 neurons were also retrogradely labeled from localized injections to the PF thalamic nucleus (*Figure 5—figure supplement 2A,B*), which is implicated in flexible and goal directed behavior (*Brown et al., 2010*; *Bradfield et al., 2013*; *Kato et al., 2018*; *Xiao et al., 2018*), and the zona incerta (ZI: *Figure 5—figure supplement 2A,B*), which subserves sensory-motor gating (*Trageser and Keller, 2004*; *Mitrofanis, 2005*). The collective targets of F2 neurons are summarized in *Figure 5F*.

F4 neurons projected to several brainstem nuclei associated with arousal and neuromodulation, including the supramammillary region (SuM), substantia nigra pars compacta (SNc), laterodorsal tegmental (LDTg), peduncular tegmental (PTg) nuclei, and the nucleus incertus (NI), as shown in *Figure 5E*. Inputs to SuM, SNc, and LDTg were confirmed with retrograde tracing experiments to originate from F4 neurons (*Figure 5—figure supplement 2C,D*). Transsynaptic labeling from the caudoventral FN, combined with immunostaining for TH, indicated that F4 neurons projection to both dopaminergic (*Figure 5E*) and non-dopaminergic SNc neurons. TH+ neurons in the ventral tegmental area (VTA), recently shown to mediate cerebellar output for social behavior (*Carta et al., 2019*), and in the retrorubral field (RRF) were also transsynaptically labeled, although fastigial axonal terminals in the VTA were very sparse (*Figure 3E* and *Figure 3—figure supplement 1D*). CHAT immunostaining demonstrated that F4 neurons project to both cholinergic and non-cholinergic neurons in the LDTg and PTg (*Figure 5E*), consistent with observations in primates (*Hazrati and Parent, 1992*). The targets of F4 neurons are summarized in *Figure 5G*.

Several regions were connected with multiple fastigial cell types. The brainstem reticular nuclei PnC, NRG, and MdV were targeted by F1$_R$, F1$_{rDLP}$, and F2 (*Figure 4—figure supplement 1*; *Figure 5—figure supplement 1A,B*). Localized anterograde tracing revealed that fastigial cell types projected to distinct subregions of these nuclei: F2 to dorsal and most medial regions, F1 to more ventral and lateral regions, and F1$_{rDLP}$ projected to dorsal and lateral regions (*Figure 4—figure supplement 1*; *Figure 5—figure supplement 1*), implying segregation of postsynaptic targets. Projections to cerebellar cortex and contralateral cerebellar nuclei by each fastigial cell may also be segregated as indicated in previous reports (*Deura, 1966*; *Buisseret-Delmas and Angaut, 1989*).

Collectively, these results establish modular connectivity patterns between the fastigial cell types and their downstream targets. Large Spp1+ F1 neurons in the rFN and rDLP connect with premotor nuclei of the caudal brainstem and spinal cord associated with posture and locomotion, and orofacial motor control, respectively. Small Snca+/Calb2+ F3 neurons in the vlFN connect specifically with neurons involved in vestibulo-autonomic and respiratory function. Medium-large Spp1+/Snca+ F2 neurons in the cDLP connect with pontine, midbrain, and spinal neurons related to orienting movements of the eyes and head and to thalamic nuclei VL and PF. Small Snca+ F4 neurons in the cFN connect with neuromodulatory nuclei of the dorsal pons and midbrain and with 'nonspecific' nuclei of the thalamus including VM, MD, and CL.

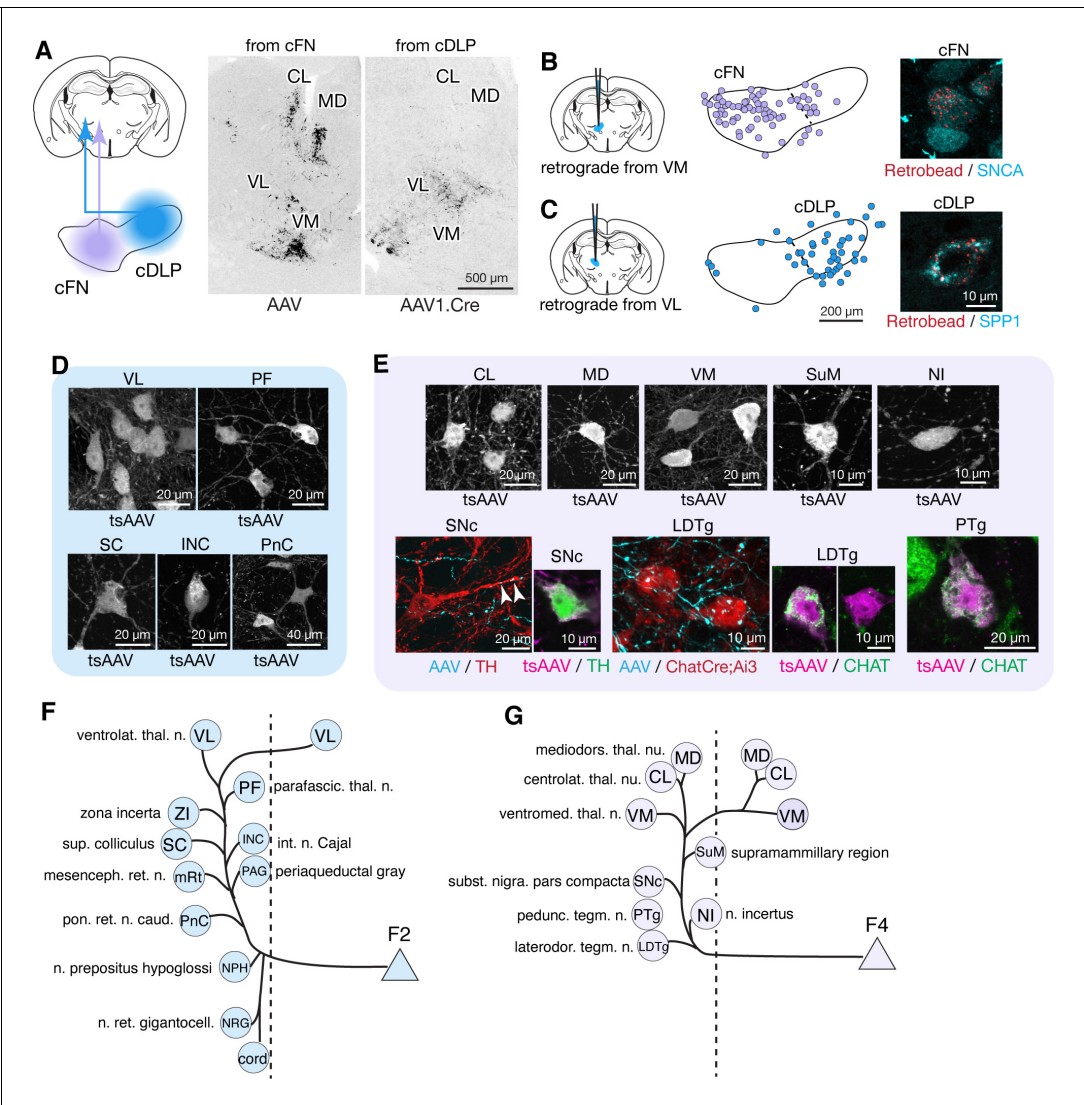

**Figure 5.** Segregated output channels from caudal parts of the FN. (**A**) AAV-mediated anterograde tracing with localized injections to the cFN and cDLP subregions of the FN revealed distinct projection targets of F2 and F4 neurons in the thalamus. A coronal thalamic section in the middle shows a case of cFN injection (territory of F4 neurons), with labeled neurons (black) in the MD, CL, and VM nuclei, but only sparsely in the VL nucleus. A similar thalamic section in the right shows a case of cDLP injection (territory of F2 neurons), with labeled neurons predominantly in the VL nucleus, but only sparsely to the MD, CL and VM nuclei. (**B and C**) Retrograde tracing experiments to confirm segregation among caudal fastigial output projections. In (**B**), retrogradely labeled neurons from the injections to the VM nucleus (n = 3 males) are mapped onto the cFN (middle); high magnification analysis shows colabeling of the retrograde tracer (retrobead, cyan) with SNCA immunoreactivity (magenta) in small cFN neurons (right), indicating that F4 neurons project to the VM nucleus. Similarly, (**C**) demonstrates that F2 neurons, which are localized in the cDLP and are SPP1+, project to the VL thalamus (n = 2 males and n = 1 female). (**D**) Anterograde transsynaptic tracing (by AAV1.hSyn.Cre, labeled as tsAAV) including the cDLP labeled neurons in the VL and PF thalamic nuclei, superior colliculus (SC), interstitial nucleus of Cajal (INC), and caudal pontine reticular nucleus (PnC) (n = 3 males and n = 2 females). (**E**) Anterograde transsynaptic tracing from the cFN labeled neurons in the CL, MD, and VM thalamic nuclei, posterior hypothalamus (PH), nucleus incertus (NI), substantia nigra pars compacta (SNc), laterodorsal tegmental nucleus (LDTg), and pedunculotegmental nucleus (PTg) (n = 3 males and n = 2 females). Fastigial axons and terminals anterogradely labeled by AAV9.RFP (labeled as AAV) contacted with TH+ SNc neuron (red) and Chat+ LDTg neurons (red), which were identified by TH immunostaining and ChatCre;Ai3 line, respectively. Anterograde transsynaptic tracing (tsAAV) confirmed the innervation of cFN neurons onto catecholaminergic neurons in the SNc (TH+, magenta) (n = 3 males and n = 1 female) and cholinergic neurons in the LDTg and PTg (CHAT+, green) (n = 3 males). (**F–G**) Summary of the major output targets of F2 (**F**) and F4 neurons (**G**). Dotted vertical line indicates midline. Targets are arranged rostro-caudally from top to bottom. Scale bars in the middle and right panels in C applies to similar panels in B. *Figure 3—source data 1* contains a complete list of projection targets. Abbreviations, cord, spinal cord; CL, centrolateral thalamic nucleus; INC, interstitial nucleus of Cajal; LDTg, laterodorsal tegmental nucleus; MD, mediodorsal thalamic nucleus; mRt, mesencephalic reticular nucleus; NPH, nucleus prepositus hypoglossi; NI, nucleus incertus; NRG, nucleus reticularis gigantocellularis; PAG, periaqueductal gray; PF, parafascicular thalamic nucleus; PH, posterior hypothalamus; PnC, pontine reticular nucleus, caudal; PTg, pedunculotegmental

*Figure 5 continued on next page*

Figure 5 continued

nucleus; SC, superior colliculus; SNc, substantia nigra, pars compacta; VL, ventrolateral thalamic nucleus; VM, ventromedial thalamic nucleus; ZI, zona incerta.

The online version of this article includes the following figure supplement(s) for figure 5:

**Figure supplement 1.** Representative results of anterograde tracing from injections localized to caudal regions of the fastigial nucleus.

**Figure supplement 2.** Analysis of retrograde tracing revealing discrete projections of F2 and F4.

## Disynaptic connections with forebrain

Nonmotor functions of the cerebellum are thought to be mediated largely by cerebellar nucleus linkages with the forebrain (*Schmahmann, 1991*; *Schmahmann, 1996*), but little is known about cell-type specific circuit connections. The substantial thalamic inputs from caudally located F2 and F4 fastigial neurons could potentially influence several subcortical and cortical areas. Accordingly, we performed AAV1- mediated transsynaptic tracing experiments to visualize axonal and synaptic terminals of fastigial-recipient thalamic neurons (*Figure 6A*, see Materials and methods).

AAV1.hSyn.Cre injected in the cFN and cDLP (F2 and F4 regions) of the fastigial nucleus of Ai14 mice produced robust transsynaptic axonal labeling in the cerebral cortex (*Figure 6B–D* and *Figure 6—figure supplement 1*). Two distinct patterns of terminal labeling were observed: widespread, diffuse labeling of layer 1 (e.g. layer 1 of secondary motor cortex in *Figure 6B*, bottom) vs localized, specific labeling of layers 3–5 (e.g. layers 3–5 of motor cortices in *Figure 6D*). Layer one terminals were evident throughout much of the cortex, including the sensorimotor, cingulate, insular, prelimbic, frontal association, lateral and ventral orbital, parietal association, and visual cortices (*Figure 6E, F* and *Figure 6—figure supplement 1*). In contrast, layer 3–5 terminals were restricted to the sensorimotor cortices (including somatosensory S1, barrel fields, and S2 and motor regions M1 and M2), frontal association cortex, and ventral/lateral orbitofrontal cortex (*Figure 6E,G* and *Figure 6—figure supplement 1*). We did not observe significant labeling in the infralimbic, medial orbitofrontal, retrosplenial, auditory or temporal cortical areas (*Figure 6—figure supplement 1*).

Disynaptic projections from the FN to subcortical forebrain regions were robust in the dorsal striatum (*Figure 6C,D,H*), consistent with a recent study using rabies virus (*Xiao et al., 2018*), with a notably dense input to the rostrolateral striatum (*Figure 6C*). Axonal terminals were also prominent in the nucleus accumbens (NAcc) of the ventral striatum (*Figure 6I*). In addition, several nuclei of the basal forebrain received disynaptic fastigial input, including the nucleus of the diagonal band of Broca (*Figure 6J*) and the medial septum, which was sparsely but consistently labeled (*Figure 6K*). In contrast with early, widely cited reports that suggested relatively direct connections from the vermis to hippocampus and amygdala (*Heath and Harper, 1974*; *Snider and Maiti, 1976*), we found no evidence for either mono- or disynaptic connections between the FN and amygdala (*Figure 6—figure supplement 1I*) or hippocampus. These results suggest that affective and cognitive functions of the vermis could be mediated by fastigial connections with the cerebral cortex (*Figure 6L*), basal ganglia (*Figure 6M*), and basal forebrain.

## Modular connections with Purkinje cells and inferior olive

The cerebellum is assumed to comprise modular circuits formed by interconnected groups of inferior olive (IO) neurons, Purkinje cells (PCs), and cerebellar or vestibular nucleus neurons (*Oscarsson, 1979*; *Ito, 1984*; *Apps and Hawkes, 2009*; *Voogd, 2011*; *Apps et al., 2018*; *Figure 7A*). Molecularly distinct groups of PCs, arranged in longitudinal stripes distinguished by the expression of Aldolase C (Aldoc, aka zebrin II; *Figure 7B,C*; *Sillitoe and Joyner, 2007*; *Cerminara et al., 2015*), are innervated by specific groups of IO neurons which, in turn, are interconnected with regions of the cerebellar nuclei innervated by of the corresponding group of PCs (*Ruigrok and Voogd, 1990*; *Ruigrok and Voogd, 2000*; *Voogd and Ruigrok, 2004*; *Sugihara and Shinoda, 2007*; *Apps and Hawkes, 2009*). To determine whether and how fastigial cell types correspond with modules comprising Purkinje cell stripes and IO neurons, we made localized injections of retrograde and anterograde tracers to FN subregions (*Figure 7D–K*).

Injections of tracers restricted to the ventral FN (F3 region) retrogradely labeled PCs confined to narrow, parasagittal bands; their mediolateral location and immunopositivity for Aldoc indicates that these PCs correspond with the 1+//1+ and 2+//3+ stripe (*Figure 7F* and *Figure 7—figure*

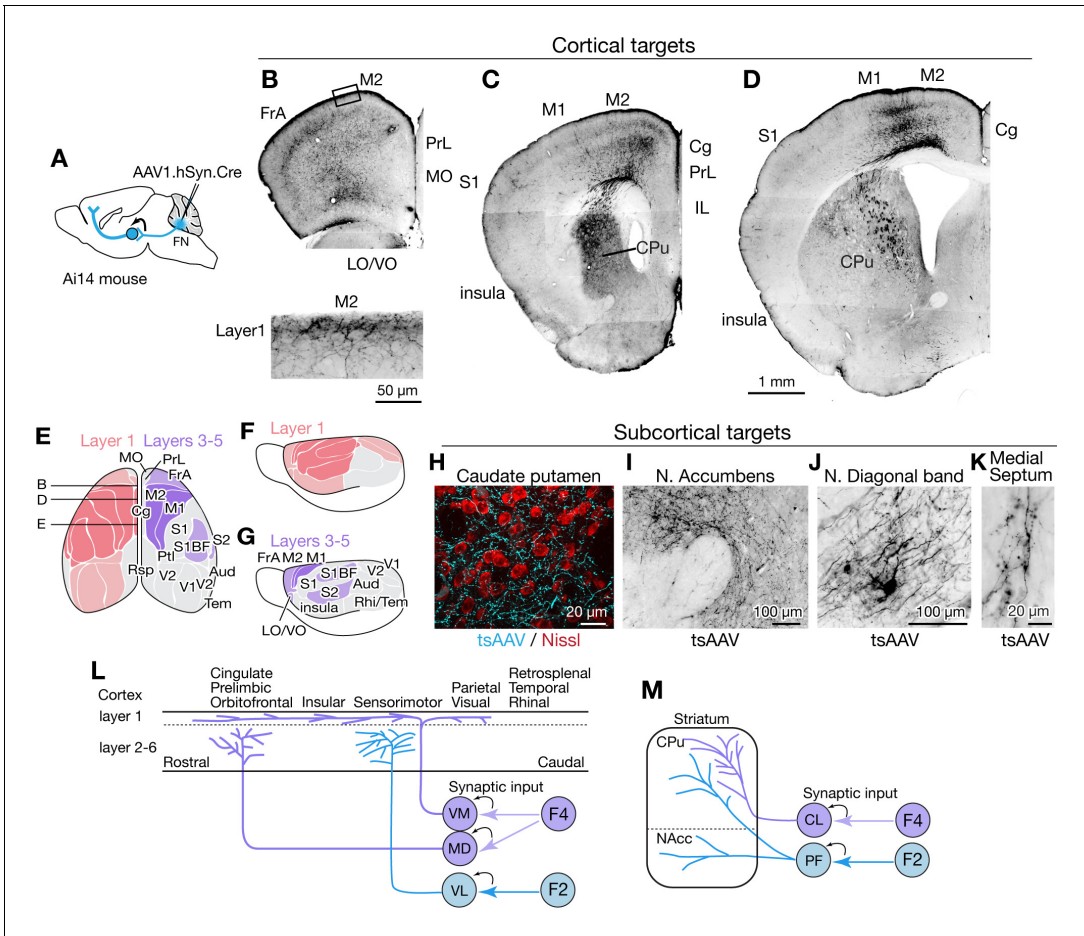

**Figure 6.** Fastigial disynaptic input to the forebrain. (**A**) Schematic of anterograde transsynaptic tracing from the FN to identify fastigial disynaptic input to forebrain areas. AAV1.hSyn.Cre injected in the FN is transsynaptically transported to postsynaptic neurons in which Cre recombinase drives reporter (tdTomato) expression in Ai14 mice, thereby axonal projections of those postsynaptic neurons to the forebrain areas are robustly labeled. (**B–D**) Disynaptic projections from the FN to the cortical areas. Panels show example coronal sections of the forebrain in which labeled axons (black) of the fastigial postsynaptic neurons show a characteristic distribution. Fastigial disynaptic input to layer one was widely distributed over the M2 (**B–D**), frontal association cortex (FrA) (**B**), lateral/ventral orbital cortex (LO/VO) (**B**), prelimbic cortex (PrL) (**B and C**), S1 (**C and D**), insular cortex (**C and D**), cingulate cortex (Cg) (**C and D**), and M1 (**D**). Inset in (**B**) is high magnification of layer 1 of the M2 (region circumscribe by rectangle in upper image), showing dense terminal arborization at layer 1. Fastigial disynaptic input also terminates at deeper layers of the FrA (**B**), LO/VO (**B**), M2 (**B–D**), and M1 (**D**). Note no labeled axons in all layers of medial orbital cortex (MO) (**B**) and infralimbic cortex (IL) (**C**), and deeper layers of insular cortex (**C**), S1 (**C and D**), and Cg (**D**). (**E–G**) Color-coded illustration of the cortical areas that receive disynaptic input from the FN to layer 1 (pink) and layers 3–5 (purple). Darker colors indicate denser input. Dorsal (**E**) and lateral (**F and G**) views are shown. Fastigial disynaptic input to layer one broadly covers the cortical areas including the FrA, PrL, LO/VO, M1, M2, S1, barrel field (S1BF), S2, insular cortex, parietal cortex (Ptl), and visual cortices (V1 and V2), but excluding MO, IL, retrosplenial cortex (Rsp), auditory cortex (Aud), and rhinal/temporal cortex (Rhi/Tem) (gray). Input to layers 3–5 (purple) is restricted in the FrA, LO/VO, M2, M1, S1BF, and S2. In (**E**), approximate levels of the sections in (**B–D**) are also shown. Results were consistent across n = 4 male and n = 4 female mice. (**H–K**) Disynaptic fastigial projection targets in subcortical forebrain areas, revealed by axonal terminals transsynaptically labeled with AAV1.hSyn.Cre (tsAAV). (**H**) The axonal terminal arborization of fastigial postsynaptic neurons (cyan) at proximity of neurons of caudate putamen (red, Nissl). (**I**) The labeled axonal terminals at the accumbens nucleus. (**J**) The labeled axonal terminals and tertiary infected neuron at the nucleus of diagonal band, a basal forebrain structure. (**K**) Sparse but consistent labeling at the medial septum. (**L**) Schematics summarizing disynaptic fastigial input to the cortex from F2 and F4 neurons. These circuits are likely to be comprised of two separate pathways, as the F2 and F4 neurons target different thalamic neurons that give rise discrete thalamocortical projections. F4 neurons provide synaptic input to the ventromedial (VM) and mediodorsal (MD) thalamic neurons. Axons of postsynaptic VM neurons target layer 1 of the widespread areas of the cortex (**Kyuhou and Kawaguchi, 1985**; **Noda and Oka, 1985**; **Kuramoto et al., 2015**). Axons of postsynaptic MD neurons target prelimbic, cingulate, and orbitofrontal cortices (**Kuramoto et al., 2017**). F2 neurons provide synaptic input to the ventrolateral (VL) thalamic neurons which then project to the deeper layers of the sensorimotor cortex (**Jones, 2009**; **Kuramoto et al., 2009**). (**M**) Schematics summarizing disynaptic fastigial input to the striatum from F2 and F4 neurons. These circuits are also likely to be comprised of two separate pathways, as the F2 and F4 neurons target different thalamic neurons that give rise discrete thalamostriatal projections. F2 neurons provide synaptic input to the parafascicular (PF) thalamic neurons that project to both the dorsal (caudate putamen; CPu) and ventral striatum (accumbens nucleus; NAcc) (**Sadikot and Rymar, 2009**). F4 neurons provide synaptic input to the

*Figure 6 continued on next page*

*Figure 6 continued*

centrolateral (CL) thalamic neurons that project to the dorsal striatum. Scale bar in D applies to low magnification images in B-D. Abbreviations, Aud, auditory cortex; Cg, cingulate cortex; CL, centrolateral thalamic nucleus; CPu, caudate putamen; FrA, frontal association cortex; IL, infralimbic cortex; LO, lateral orbital cortex; M1 and M2, primary and secondary motor cortex; MD, mediodorsal thalamic nucleus; MO, medial orbital cortex;; PF, parafascicular thalamic nucleus; PrL, prelimbic cortex; Ptl, parietal association cortex; Rhi, ecto-/peri-/ento-entorhinal cortex; Rsp, retrosplenial cortex; S1 and S2, primary and secondary sensory cortex; S1BF, barrel field of primary sensory cortex; Tem, temporal association cortex; V1 and V2, primary and secondary visual cortex; VL, ventrolateral thalamic nucleus; VM, ventromedial thalamic nucleus; VO, ventral orbital cortex.

The online version of this article includes the following figure supplement(s) for figure 6:

**Figure supplement 1.** Overview of disynaptic fastigial input to the forebrain.

*supplement 1B*; see Methods for stripe nomenclature). In contrast, PCs located between these bands, in the 1-//1-/2- stripe, were retrogradely labeled from rFN (F1$_R$ region) (*Figure 7G* and *Figure 7—figure supplement 1A*). rDLP injections (F1$_{rDLP}$ region) retrogradely labeled PCs in the para-vermal regions of the simple lobule and Crus II, which were immunonegative for Aldoc and thus identified as 2b-//4a- and c-//4b- stripes (*Figure 7H*). Injections into the caudalmost portion of the FN, including the cFN (F4 region) and cDLP (F2 region), retrogradely labeled PCs that were restricted to the a+//2+ stripe in vermis lobules VI-IX and to the c+//4b+ stripe in Crus I (*Figure 7I*). Injections of AAV1.hSyn.Cre to label PCs in the medial part of Crus I (c+//4b+ region) of Ai14 mice transsynaptically labeled F2 neurons in the cDLP (*Figure 7L*). In contrast, PC axonal terminals in the F4 region of the cFN were labeled by Cre-dependent AAV injections to lobule VIII in L7Cre mice (*Figure 7L*). These connectivity patterns are consistent with results from single axon tracing studies in rats (*Sugihara et al., 2009*).

Complementary anterograde tracing experiments revealed that fastigial connections with IO subnuclei are also organized modularly. Pan-fastigial injections of AAV1.hSyn.Cre to Ai14 mice resulted in transsynaptic neuronal labeling throughout the contralateral caudal medial accessory olive (cMAO) and neighboring beta nucleus (*Figure 7D*). Each subdivision of the caudal IO was transsynaptically labeled from a specific subregion of the FN; injections in the vlFN, rFN and rDLP labeled neurons in the subnuclei a, b, and c of the cMAO, respectively (*Figure 7F–H*). Notably, climbing fibers from cMAO subnuclei a, b, and c innervate PCs associated with the vlFN (F3 region), rFN (F1$_R$ region), and rDLP (F1$_{rDLP}$ region), respectively (*Sugihara and Shinoda, 2004*). Injections in the cFN transsynaptically labeled neurons in the beta subdivision of the IO (*Figure 7K*), which project to PCs that innervate the cFN (F4 region) (*Sugihara and Shinoda, 2004*). Injections that included both the cFN and cDLP additionally labeled neurons interposed between beta and subnucleus c which we have termed subnucleus 'd' (*Figure 7J*, *Figure 7—figure supplement 2*, see Methods). IO neurons in subnucleus d innervate PCs in lobules VIc/VII and Cr I (*Figure 7—figure supplement 2E,F*) which are linked with cDLP (F2) (*Sugihara et al., 2009*; *Sarpong et al., 2018*). Collectively, these results demonstrate that the excitatory fastigial cell types identified in this study are each linked with a specific group of interconnected PCs and IO neurons (*Figure 7M*).

## Discussion

This study demonstrated that medial cerebellar outputs are mediated by molecularly distinct fastigial cell types which are modularly connected with specific sets of Purkinje cells, inferior olive neurons, and downstream targets (*Figure 8*). We identified five major types of glutamatergic projection neurons, which can be divided into two classes. Large F1$_R$ and F1$_{rDLP}$ neurons, innervated by Aldoc-negative PCs, connect primarily with large brainstem neurons associated with online control of motor and autonomic functions. In contrast, smaller F2, F3, and F4 neurons, innervated by Aldoc-positive PCs, connect with multiple downstream circuits associated with sensory processing, motor preparation and arousal. Each excitatory fastigial cell type receives convergent inputs from widely distributed Purkinje cells associated with a specific set of IO neurons and makes divergent projections to a functionally related collection of downstream targets. These results suggest that diverse functions of the cerebellar vermis are mediated by the parallel operation of several discrete multineuronal circuit modules. Specific disynaptic linkages with prefrontal cortex, striatum, basal forebrain, and brainstem arousal circuits made by caudal fastigial neurons (F2 and F4) could underlie the cognitive and

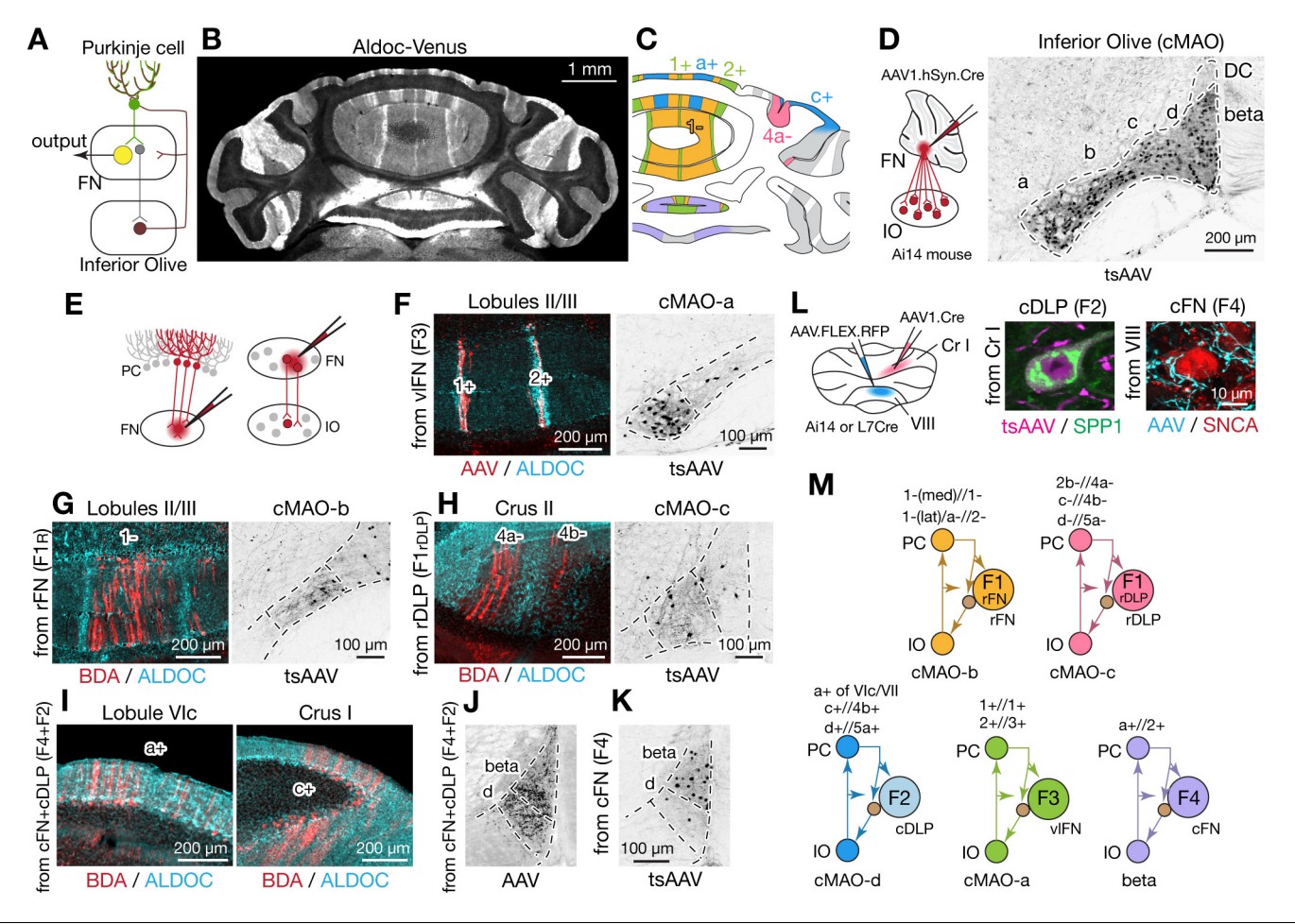

**Figure 7.** Modular connectivity of the fastigial subregions with specific PC and IO neurons. (A) Schematic illustrating circuit connections within the canonical olivo-cerebellar loop. Neurons in a subnucleus of inferior olive (IO, garnet) innervate a specific set of Purkinje cells (PCs, green) in the cerebellar cortex, as well as their target subregions in the cerebellar nuclei. GABAergic neurons in the cerebellar nuclei (grey) in turn project back to the original IO subnucleus, closing the circuit loop between the IO and cerebellum. In the cerebellar nuclei, excitatory output neurons (yellow) and GABAergic nucleoolivary neurons which share presynaptic Purkinje cells reside in the same subregions. (B) Cerebellar coronal section from the Aldoc-venus mouse demonstrates alternating stripes of Aldoc-positive and -negative PCs. (C) PC stripes shown in (B), color-coded to indicate connections with specific fastigial cell types. Stripe terminology corresponds with *Sugihara and Shinoda, 2004*. (D) IO neurons that are postsynaptic to the FN. Schematic illustrates anterograde transsynaptic pan-FN injections of AAV1.hSyn.Cre (tsAAV) to label postsynaptic IO neurons. Right image shows clusters of labeled IO neurons (black) in subnuclei a-d of the caudal medial accessory olive (cMAO-a, b, c, and d) and the beta subnucleus, with no neurons labeled in the dorsal cap (DC) (n = 5 males and n = 3 females). (E) Schematics illustrating localized retrograde and anterograde tracer injections into the FN to identify presynaptic PCs and postsynaptic IO neurons, respectively. (F–H) Retrogradely labeled PCs and anterogradely and transsynaptically labeled IO neurons from localized tracer injections to vlFN, rFN, and rDLP. (F) Shows results of retrograde infection of AAV1.Cre injected into F3 region (n = 2 males), in which labeled PCs with AAV1.Cre (red) are located in the 1+ and 2+ stripes and are immunopositive for ALDOC (cyan; left panel); transsynaptically labeled IO neurons (black, right panel) from the same injections are located in cMAO-a (n = 2 males). Similarly, (G) Shows results of BDA injections into rFN (F1_R region; n = 2 males and n = 1 female), in which labeled PCs with BDA (red) are located at 1- stripe and are immunonegative for ALDOC (cyan, left panel); transsynaptically labeled IO neurons from injections of AAV1.Cre into rFN are located at cMAO-b (black, right panel) (n = 3 females). (H) Shows results of BDA injections into rDLP (F1_{rDLP} region; n = 2 males), in which labeled PCs with BDA (red) are located at 4a-/4b- stripes and are immunonegative for ALDOC (cyan) (left panel); transsynaptically labeled IO neurons from injections of AAV1.Cre into rDLP are located at cMAO-c (black, right panel) (n = 3 females). (I) Retrogradely labeled PCs from BDA injections that hit both cFN and cDLP (F4 and F2 regions). Labeled PCs (red) are located in the a+ and c+ stripes and are immunopositive for ALDOC (cyan) (n = 4 males and n = 2 females). (J) Anterogradely labeled fastigial axons innervating cMAO-d and beta from an injection that included both the cFN and cDLP (n = 6 males and n = 2 females). (K) Transsynaptically labeled beta neurons from an injection that hit cFN but not cDLP (n = 2 males). (L) Anterograde tracing experiments to identify PC projections to caudal portions of FN. Left schematic illustrates injection experiments, in which AAV1.hSyn.Cre was injected to the Crus I (magenta) of Ai14 mice to transsynaptically label FN neurons (n = 2 males), and Cre-dependent AAV, AAV.FLEX.RFP, was injected to lobule VIII (cyan) of L7Cre mice to label PC axons innervating the FN (n = 2 males). In the middle panel, transsynaptically labeled neurons (magenta) from the Crus I

*Figure 7 continued on next page*

Figure 7 continued

injections labeled neurons in cDLP confirmed as F2 by their medium size and SPP1-immunopositivity. In the right panel, PC terminals (cyan) from injections in lobule VIII contacted SNCA+ (red) small neurons in the cFN (F4 neurons). (M) Diagrams schematize five olivo-cerebellar-fastigial loops identified in this study. Arrows from brown circles indicate GABAergic nucleoolivary projections. Specific olivocerebellar projections, Purkinje cell projections, and nomenclature of the cerebellar cortical stripes are based on *Sugihara and Shinoda, 2004*, *Sugihara and Shinoda, 2007*; *Voogd and Ruigrok, 2004*; *Sugihara et al., 2009*. Abbreviations, BDA, biotinylated dextran amine; cMAO-a, b, c, and d, caudal medial accessory olive, subnucleus a, b, c, and d; Cr I, crus I; Cr II, crus II; DC, dorsal cap of Kooy; FN, fastigial nucleus; IO, inferior olive; PC, Purkinje cell.

The online version of this article includes the following figure supplement(s) for figure 7:

**Figure supplement 1.** Convergent Purkinje cell projections from multiple lobules to fastigial subregions.

**Figure supplement 2.** Distinct input and output projections of the caudal medial accessory olive subnuclei 'd' and c.

affective disturbances associated with vermal malfunction (*Courchesne et al., 1988*; *Schmahmann and Sherman, 1998*; *Badura et al., 2018*).

## Diversity of medial cerebellar projection neurons

Although the fastigial nucleus was known to comprise multiple cell types (*Flood and Jansen, 1961*; *Matsushita and Iwahori, 1971*; *Beitz and Chan-Palay, 1979*; *Chen and Hillman, 1993*; *Ikeda et al., 1995*; *Bagnall et al., 2009*; *Chung et al., 2009*), it has been broadly assumed that outputs are mediated predominantly by large excitatory neurons. Our findings demonstrate multiple subtypes of glutamatergic projection neurons which vary in size, input-output connectivity, and expression levels of genes associated with fast firing and rapid signal transmission (*Kodama et al., 2020*). An additional population of large glycinergic neurons in the rostral fastigial nucleus project ipsilaterally to brainstem nuclei targeted by contralateral F1 excitatory fastigial neurons (*Bagnall et al., 2009*). GABAergic nucleo-olivary neurons, abundant in the other cerebellar nuclei (*Prekop et al., 2018*), are relatively sparsely distributed in each subregion of the fastigial nucleus (*Ruigrok and Teune, 2014*). Interestingly, as with fastigial F3 and F4 neurons, small glutamatergic neurons in the dentate (lateral) cerebellar nucleus are located ventrally and are associated with nonmotor functions of the cerebellum (*Dum et al., 2002*; *Küper et al., 2011*; *Ashmore and Sommer, 2013*; *Kunimatsu et al., 2016*).

The principle that the nervous system minimizes axonal caliber and energy usage (*Attwell and Laughlin, 2001*; *Perge et al., 2012*; *Stevens, 2012*) suggests that large, fast-spiking cerebellar nucleus neurons may be specialized to meet evolutionarily critical demands. Purkinje cells with the fastest firing capacity (Aldoc-negative: *Xiao et al., 2014*; *Zhou et al., 2014*) densely innervate the largest fastigial neurons (F1$_R$), which in turn make extensive synapses onto somata and proximal dendrites of gigantic brainstem neurons responsible for rapid posturomotor reflexes (*Eccles et al., 1975a*). The largest glutamatergic fastigial cell types can transform Purkinje cell synaptic inhibition into precisely timed spikes (*Özcan et al., 2020*), as has been shown for their counterparts in the interpositus and dentate nuclei (*Person and Raman, 2012*). On the other hand, physiological properties of the smallest glutamatergic fastigial cell types (F3 and F4) have not been explicitly reported, likely reflecting challenges in isolating and targeting them for recordings. However, their small size and lower expression of genes related to fast spiking (*Kodama et al., 2020*) suggest that their responses to synaptic inputs may be slower and less time-locked than those of larger cell types. We conjecture that cell size differences across fastigial projection neurons reflect their distinct temporal operating domains as discussed below: fast online control of axial and orofacial actions by the largest cells (F1$_R$ and F1$_{rDLP}$), orienting responses by intermediately sized cells (F2), and tonic modulation of autonomic functions and arousal by the smallest cells (F3 and F4).

## Cerebellar fastigial circuit modules

Although it has long been appreciated that fastigial nucleus neurons target widespread subcortical regions, with rostral vs caudal fastigial neurons projecting predominantly to hindbrain vs midbrain and thalamus (*Angaut and Bowsher, 1970*; *Batton et al., 1977*; *Bentivoglio and Kuypers, 1982*; *Teune et al., 2000*; *Bagnall et al., 2009*), the input-output linkages made by specific fastigial cell types have not been previously identified. Our results demonstrate that each major excitatory fastigial cell type is differentially linked with a specific set of inferior olive neurons, Purkinje cells, and functionally related downstream targets (*Figure 8*). This remarkable circuit modularity provides a

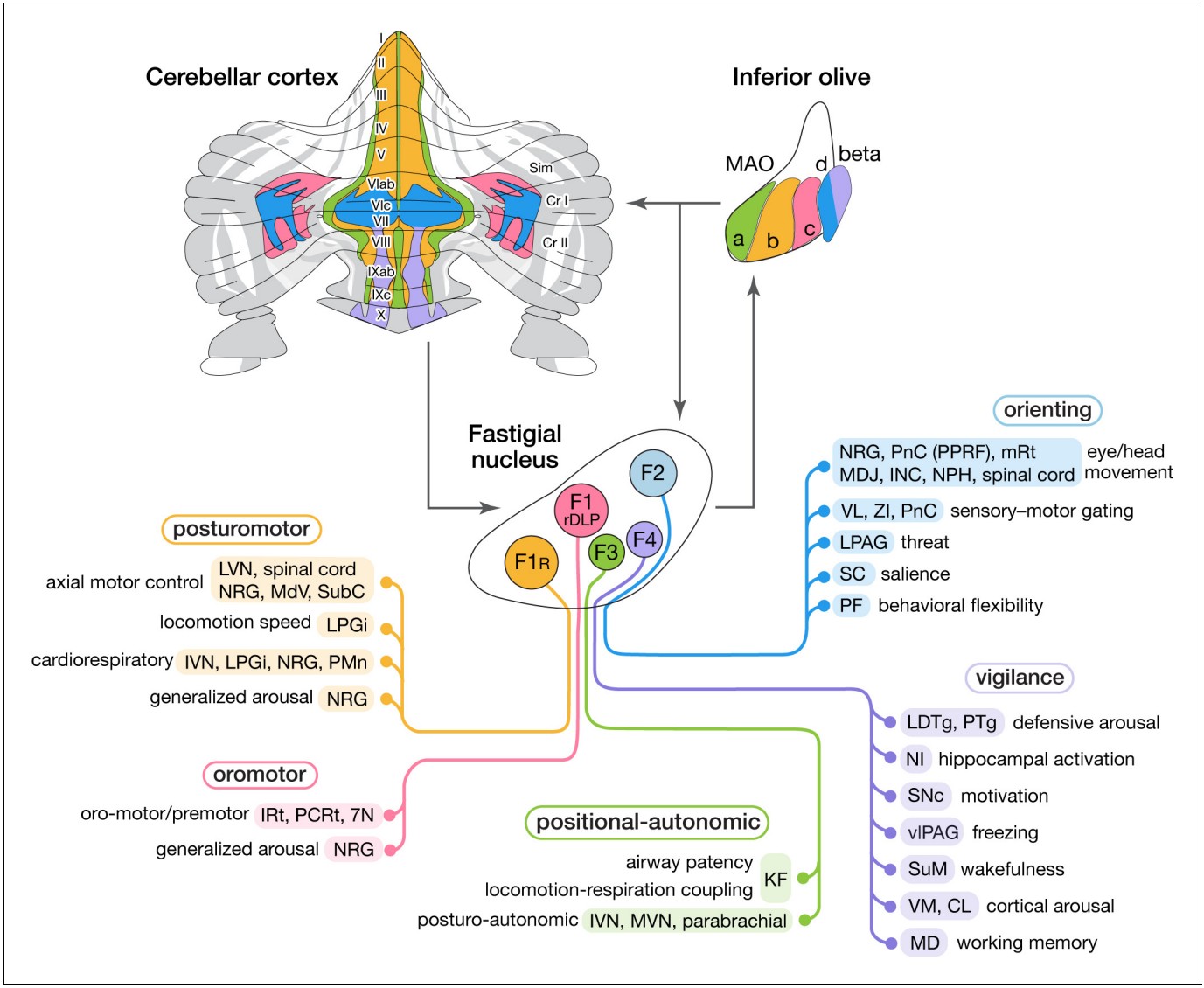

**Figure 8.** Modular circuit connections of excitatory fastigial projection neurons provide circuit substrates for coordinating five broad organismal functions. Schematics summarize cerebellar modular circuit connections that link distinct types of fastigial nucleus neurons with specific neurons in the inferior olive, cerebellar cortex, and downstream brain regions. Projection targets of each FN cell type are indicated in shaded colors. Specific functions associated with each collection of projection targets are indicated at the left; proposed broad organismal functions of each module are encircled above. To show the distribution of Purkinje cells associated with each module, a flatmap of the mouse cerebellar cortex with vermis lobules indicated numerically and Aldoc/zebrin stripes indicated in grey, with medial-lateral width expanded 5x for clarity, was redrawn from *Fujita et al., 2014*. Inferior olive subnuclei of the caudal MAO are denoted a, b, c, d and beta. Circles around each fastigial nucleus cell type indicate relative size and parasagittal position. Abbreviations, 7N, facial nucleus; CL, centrolateral thalamic nucleus; Cr I, Crus I; Cr II, Crus II; INC, interstitial nucleus of Cajal; IRt, intermediate reticular nucleus; IVN, inferior vestibular nucleus; KF, Kölliker-Fuse nucleus; LDTg, laterodorsal tegmental nucleus; LPAG, lateral periaqueductal gray; LPGi, lateral paragigantocellular nucleus; LVN, lateral vestibular nucleus; MAO, medial accessory olive; MD, mediodorsal thalamic nucleus; MDJ, mesodiencephalic junction; MdV, medullary reticular nucleus, ventral; mRt, mesencephalic reticular formation; MVN, medial vestibular nucleus; NPH, nucleus prepositus hypoglossi; NI, nucleus incertus; NRG, nucleus reticularis gigantocellularis; PCRt, parvocellular reticular nucleus; PF, parafascicular thalamic nucleus; PMn, paramedian reticular nucleus; PnC, pontine reticular nucleus, caudal; PPRF, paramedian pontine reticular formation; PTg, pedunculotegmental nucleus; Sim, simplex lobule; SC, superior colliculus; SNc, substantia nigra, pars compacta; SubC, subcoeruleus nucleus; SuM, supramammillary region; VL, ventrolateral thalamic nucleus; vlPAG, ventrolateral periaqueductal gray; VM, ventromedial thalamic nucleus; ZI, zona incerta.

heuristic framework for considering the diverse functions associated with the cerebellar vermis and fastigial nucleus, as described below.

The $F1_R$ module includes brainstem and spinal nuclei associated with axial motor control (*Eccles et al., 1975a*; *Mori et al., 2001*; *Lois et al., 2009*), locomotion speed (*Capelli et al., 2017*), cardiorespiratory control (*Miura and Reis, 1971*; *Xu and Frazier, 1995*), and generalized arousal (*Pfaff et al., 2012*). Vermis PCs in the $F1_R$ module respond to proprioceptive, somatosensory, and vestibular signals related to self-motion (*Oscarsson and Sjölund, 1977*; *Pompeiano et al., 1997*; *Manzoni et al., 1999*; *Muzzu et al., 2018*). The corresponding IO neurons in the subnucleus b of the cMAO convey error signals from cutaneous and proprioceptive spino-olivary afferents from the trunk and hindlimb (*Oscarsson and Sjölund, 1977*; *Gellman et al., 1983*), including group III afferents, which encode muscle fatigue (*Amann, 2012*). Collectively, these circuit connections could enable $F1_R$ neurons, together with their ipsilaterally projecting glycinergic counterparts (*Bagnall et al., 2009*), to adaptively coordinate somatomotor, autonomic, and arousal functions during upright stance, changes in posture, and locomotion.

The $F1_{rDLP}$ module includes medullary nuclei associated with orofacial motor control (*Lu et al., 2013*; *Stanek et al., 2014*; *McElvain et al., 2018*), respiratory modulation (*Xu and Frazier, 2002*; *Lu et al., 2013*; *Moore et al., 2013*), and arousal (*Pfaff et al., 2012*), consistent with responses of the DLP neurons to signals related to whisking, breathing, and licking (*Lu et al., 2013*). PCs in the $F1_{rDLP}$ module, located in medial parts of the simple lobule and Crus II, receive orofacial tactile signals (*Bower and Woolston, 1983*), and fire action potentials time-locked to licking (*Welsh et al., 1995*) and whisking (*Brown and Raman, 2018*), as do the associated molecular layer interneurons (*Astorga et al., 2017*). Inferior olive neurons in the $F1_{rDLP}$ module (cMAO-c) respond to perioral stimuli driven by indirect trigeminal inputs from lateral superior colliculus (*Akaike, 1988*) and the mesodiencephalic junction (*Kubo et al., 2018*). These circuit connections make the $F1_{rDLP}$ module well suited for contextual modulation of trigemino-motor reflexes and the adaptive coordination of multiple oromotor behaviors and breathing.

The F3 module includes caudal vestibular nuclei (IVN and MVN), which mediate postural modulation of autonomic functions (*Yates, 1996*; *Yates et al., 2014*), the Kölliker Fuse nucleus, which modifies sensory responsiveness of brainstem respiratory circuits (*Dutschmann and Herbert, 2006*), and the parabrachial complex, thought to be critical for arousal responses to homeostatic distress (*Palmiter, 2018*). Notably, localized stimulation of the ventral fastigial nucleus evokes increases in blood pressure (*Doba and Reis, 1972*), and chemosensitive neurons responsive to $CO_2$ are similarly localized ventrally (*Martino et al., 2007*), consistent with a role for F3 fastigial neurons in cardiovascular and respiratory responses to head tilt (*Woodring et al., 1997*; *Yates et al., 2002*; *Wilson et al., 2006*) and hypercapnia (*Xu and Frazier, 2002*). PCs in the F3 module extend over cerebellar cortical midline regions that receive vestibular and proprioceptive signals related to head and body position with respect to gravity (*Laurens et al., 2013*; *Luo et al., 2020*). The F3 region of the medial accessory olive (cMAO-a) receives inputs from the lumbosacral spinal cord (*Molinari, 1985*; *Matsushita et al., 1992*) and from cardiorespiratory regions of the nucleus tractus solitarius (*Loewy and Burton, 1978*). These circuit connections suggest that the F3 module contributes to adaptive adjustments of autonomic reflexes during changes in body position. The F3 module, together with nodulus/uvula connections with vestibular and parabrachial nuclei (*Nisimaru, 2004*; *Sugihara et al., 2009*) may be critical for postural regulation of blood pressure and airway patency; dysfunction of these circuits could account for the cerebellar links with orthostatic hypotension (*Rector et al., 2006*) and sleep-disordered breathing (*Harper et al., 2015*).

The F2 module encompasses several brainstem, midbrain, and thalamic nuclei that subserve distinct aspects of the orienting response to novel stimuli (*Pavlov, 1927*). Projection targets of F2 neurons are remarkably congruent with those of the primate fastigial oculomotor region (*Noda et al., 1990*) and are related to eye and head movement (*Isa and Sasaki, 2002*; *Takahashi et al., 2014*), threat processing (*Wager et al., 2009*; *Faull and Pattinson, 2017*), sensory gating (*Massion, 1976*; *Fendt et al., 2001*; *Trageser and Keller, 2004*; *Mitrofanis, 2005*), stimulus salience (*Knudsen, 2018*), and behavioral flexibility (*Brown et al., 2010*; *Bradfield et al., 2013*; *Kato et al., 2018*). PCs in the F2 module, located in lobules VIc and VII (aka 'oculomotor vermis': *Ohtsuka and Noda, 1991*) and medial Crus I, are downstream of pontine mossy fibers (*Päällysaho et al., 1991*; *Biswas et al., 2019*) which convey signals from superior colliculus (*Ohtsuka and Noda, 1992*; *Kojima and Soetedjo, 2018*), motor, premotor, cingulate, and prelimbic cortices (*Watson et al.,*

*2009*; *Coffman et al., 2011*), and subthalamic nucleus (*Jwair et al., 2017*). IO neurons in the F2 module (cMAO-d) receive inputs from the medial superior colliculus and frontal cortical regions (*Akaike, 1992*; *Watson et al., 2009*; *Voogd and Barmack, 2006*). Circuit connections with superior colliculus, basal ganglia, frontal cortex, and premotor brainstem could enable the F2 module to coordinate behavioral, cognitive, and arousal responses to novel or unexpected events. Accordingly, dysfunction of the F2 module could impair both motoric and nonmotoric aspects of orienting to environmental stimuli.

The F4 module includes pontine, midbrain, and thalamic nuclei implicated in arousal and vigilance, consistent with classical reports of fastigial contributions to the ascending reticular activating system (*Moruzzi and Magoun, 1949*) and to regulation of affect (*Schmahmann and Sherman, 1998*). Robust connections with cholinergic and noncholinergic neurons in the LDTg and PTg, which are associated with vigilance and arousal (*Steriade et al., 1990*; *Reese et al., 1995*; *Lee and Dan, 2012*), have not been previously reported, possibly reflecting restricted tracer uptake or selective cytotoxicity of small F4 neurons. Connections with the nucleus incertus, supramammillary region, and VM thalamic nuclei enable F4 neurons to coordinate activity of the hippocampus (*Martínez-Bellver et al., 2017*), basal forebrain (*Brown and McKenna, 2015*; *Pedersen et al., 2017*), and broad regions of the cerebral cortex (*Steriade, 1995*; *Ma et al., 2017*; *Honjoh et al., 2018*). PCs in the F4 module are located in posterior vermal lobules associated with affective dysregulation (*Stoodley et al., 2017*; *Badura et al., 2018*), pain, and addiction (*Moulton et al., 2010*; *Moulton et al., 2014*). IO beta neurons in the F4 module receive inputs from medial frontal cortex (*Swenson et al., 1989*) and from vestibular neurons which encode head tilt, pitch, and translation (*Barmack et al., 1993*; *Ma et al., 2013*). Consistent with a role in signaling changes in arousal level, IO beta neurons are activated by anesthesia (*Rutherfurd et al., 1992*; *Lantéri-Minet et al., 1993*), vestibular stimuli (*Barmack et al., 1993*; *Ma et al., 2013*), vagal nerve stimulation (*Yousfi-Malki and Puizillout, 1994*), and drug-induced hyperactivity (*Näkki et al., 1996*). Accordingly, the E4 module could provide a widely broadcasted alerting signal that coordinately engages the basal forebrain, hippocampus, and cerebral cortex under conditions that require vigilant arousal.

## Technical considerations

The small size of the fastigial nucleus and the overlapping trajectory of axonal projections from distinct cell types made it technically challenging to identify the postsynaptic targets associated with each cell type. When used in isolation, circuit tracing techniques can each produce ambiguous results which are subject to misinterpretation. Retrograde tracers can be taken up by axons of passage (*Lanciego and Wouterlood, 2011*; *Saleeba et al., 2019*), some 'anterograde' tracers can be transported in the wrong direction (*Lo and Anderson, 2011*; *Rothermel et al., 2013*; *Zingg et al., 2020*), many types of tracers suffer from cell type selective infectivity (*Nassi et al., 2015*), and confining injections to small structures can be difficult. We relied on a combination of retrograde and anterograde tracing, careful assessment of a large number of cases, and strict inclusion criteria for injection sites to rigorously disambiguate circuit connections.

The AAV1-mediated transsynaptic tracing method applied in this study resulted in robust labeling of neurons in brain regions innervated by fastigial terminals. We consider these neurons to be postsynaptic to the fastigial nucleus given that synaptic release mechanisms mediate the transsynaptic transport (*Zingg et al., 2020*). A few limitations, however, should be noted. First, tropism of this AAV for certain cell-types can over- or under-estimate the synaptic connectivity. For example, transsynaptic transport of this virus to neuromodulatory neurons is ineffective (*Zingg et al., 2020*), consistent with our observations of sparse postsynaptic labeling in the PAG (*Figure 3E*), where the majority of the fastigial projections terminate on dopaminergic cells (*Vaaga et al., 2020*). AAV tropism (*Nassi et al., 2015*) should also be considered when interpreting disynaptic fastigial input to the forebrain, as some cell groups in the thalamus might be more efficiently infected by this virus. Relatedly, rabies virus (e.g. *Schwarz et al., 2015*) also has tropism limitations (*Reardon et al., 2016*; *Ruigrok et al., 2016*; *Luo et al., 2018*; *Beier, 2019*); indeed, robust fastigial or vermis connection with the forebrain were not identified in previous rabies tracing studies (*Kelly and Strick, 2003*; *Hoshi et al., 2005*). Second, the retrograde infectivity of the AAV (*Rothermel et al., 2013*; *Zingg et al., 2020*) should be considered when interpreting results. We took advantage of this to retrogradely identify PCs that project to specific fastigial subregions. Third, we have observed occasional labeling of the presumed third order neurons (e.g. pyramidal neurons; *Figure 6—figure*

*supplement 1F*), in contrast with the original description of the method (*Zingg et al., 2017*), in which the transsynaptic transport was reported to be exclusively monosynaptic. Assuming that the third order labeling is a result of our 10-fold more copies of injections than the original report, this method may require specific titers for the transsynaptic labeling to indicate monosynaptic vs disynaptic transport.

### Cerebellar modules vs lobules

Although cerebellar functions have typically been linked with lobules (*MacKay and Murphy, 1979*; *Welker, 1990*; *Stoodley and Schmahmann, 2009*; *Badura et al., 2018*; *Heffley and Hull, 2019*), we demonstrate that cerebellar outputs are organized in modules which extend across multiple lobules. For example, vermis lobule VIII is a composite of three cerebellar modules which are differentially linked with brainstem nuclei subserving somatomotor, autonomic, and arousal functions (F1$_R$, F3, and F4; see *Figure 8*), consistent with the mixed motor and mon-motor representations of this lobule identified in human imaging studies (*Stoodley et al., 2012*; *E et al., 2014*; *Guell et al., 2018a*). Similarly, each of the vermal lobules I-IX comprises multiple modules (*Figure 8*). Differences in the combinations and proportions of individual modules across lobules are likely to underlie different aspects of the same broad functions distributed across lobules, such as fear-learning by lobule VI/V vs freezing by lobule VIII (*Apps et al., 2018*). The parasagittal distribution of PCs in each module contrasts with the transverse organization of parallel fibers, enabling each FN cell type to integrate signals from multiple types of mossy fibers. Given that many of the FN projection targets are positioned to serve as circuit hubs which coordinate complex functions by distributing signals to a wide set of postsynaptic neurons, each fastigial cell type can be thought of as a superhub. Multiple modules are likely to operate in parallel to subserve broad organismal functions (*Guell et al., 2018b*; *Diedrichsen et al., 2019*; *King et al., 2019*); parallel fibers within each lobule could serve to connect functionally related modules (*Valera et al., 2016*).

### Nonmotor circuits of the cerebellar vermis and fastigial nucleus

In contrast to the widely accepted linkages of the cerebellar hemispheres and dentate nucleus with prefrontal cortex and cognitive functions, whether the vermis and fastigial nucleus participate in nonmotor functions has been controversial (*Glickstein, 2007*; *Gao et al., 2018*; *De Schutter, 2019*; *Gao et al., 2019*). By using a sensitive AAV-mediated anterograde transsynaptic tracing methods, we identified robust disynaptic fastigial nucleus connections with several regions of prefrontal cortex (*Steriade, 1995*; *Watson et al., 2009*; *Watson et al., 2014*; *Badura et al., 2018*; *Kelly et al., 2020*) mediated predominantly by small, ventrally located F4 neurons. Consistent with long standing physiological findings that stimulation of the FN evokes prominent changes in electroencephalographic activity throughout much of the cerebral cortex (*Dow and Moruzzi, 1958*; *Manzoni et al., 1967*; *Steriade, 1995*), we demonstrated strikingly dense disynaptic input from the FN to widespread regions of the forebrain, including the cortex, striatum, and basal forebrain. Modular connections of fastigial neurons with brainstem arousal, autonomic, and medial thalamic nuclei provide candidate circuit substrates for the cerebellar cognitive affective syndrome (*Schmahmann and Sherman, 1998*; *Albazron et al., 2019*) and could account for the therapeutic effect of vermis intervention on neuropsychiatric disorders (*Demirtas-Tatlidede et al., 2010*; *Parker et al., 2014*; *Stoodley et al., 2017*; *Brady et al., 2019*; *Kelly et al., 2020*).

## Materials and methods

**Key resources table**

| Reagent type (species) or resource | Designation | Source or reference | Identifiers | Additional information |
|---|---|---|---|---|
| Strain, strain background (*Mus musculus*) | C57BL/6J | Jackson Laboratory | stock 000664 | |

*Continued on next page*

*Continued*

| Reagent type (species) or resource | Designation | Source or reference | Identifiers | Additional information |
|---|---|---|---|---|
| Genetic reagent (*Mus musculus*) | YFP-16; Tg (Thy1-YFP)16Jrs | Dr. Joshua R Sanes | MGI:3505585 | PMID:11086982 |
| Genetic reagent (*Mus musculus*) | GlyT2-EGFP; Tg(Slc6a5-EGFP)1Uze | Dr. Hanns Ulrich Zeilhofer | MGI:3835459 | PMID:15611994 |
| Genetic reagent (*Mus musculus*) | Gad2-nls-mCherry; STOCK Gad2$^{tm2(cre)Zjh}$/J ; Gad2$^{2A-mCherry}$ | Jackson Laboratory | stock 023140 | PMID:25913859 |
| Genetic reagent (*Mus musculus*) | Lhx6-EGFP; Tg(Lhx6-EGFP) BP221Gsat | MMRRC | stock 000246 | PMID:14586460 |
| Genetic reagent (*Mus musculus*) | L7Cre; B6.129-Tg(Pcp2-cre)2Mpin/J | Jackson Laboratory | stock 004146 | PMID:11105049 |
| Genetic reagent (*Mus musculus*) | ChatCre; B6N.129S6(B6)-Chat$^{tm2(cre)Lowl}$/J | Jackson Laboratory | stock 018957 | PMID:21284986 |
| Genetic reagent (*Mus musculus*) | Gad2Cre; B6N.Cg-Gad2$^{tm2(cre)Zjh}$/J | Jackson Laboratory | stock 019022 | |
| Genetic reagent (*Mus musculus*) | VgluT2-Cre; STOCK Slc17a6$^{tm2(cre)Lowl}$/J | Jackson Laboratory | stock 016963 | PMID:21745644 |
| Genetic reagent (*Mus musculus*) | Ai3; B6.Cg-Gt(ROSA) 26Sor$^{tm3(CAG-EYFP)Hze}$/J | Jackson Laboratory | stock 007903 | PMID:22446880 |
| Genetic reagent (*Mus musculus*) | Ai14; B6;129S6-Gt(ROSA)26Sor$^{tm14(CAG-tdTomato)Hze}$/J | Jackson Laboratory | stock 007908 | PMID:22446880 |
| Genetic reagent (*Mus musculus*) | Ai32; B6.Cg-Gt(ROSA)26Sor$^{tm32(CAG-COP4*H134R/EYFP)Hze}$/J | Jackson Laboratory | stock 024109 | PMID:22446880 |
| Genetic reagent (*Mus musculus*) | Sun1-sfGFP; B6;129-Gt(ROSA)26Sor$^{tm5(CAG-Sun1/sfGFP)Nat}$/J | Jackson Laboratory | stock 021039 | PMID:26087164 |
| Antibody | anti-alpha synuclein (SNCA), mouse monoclonal | BD Bioscience | Cat# 610786, RRID:AB_398107 | dilution 1:100 |
| Antibody | anti-osteopontin (SPP1), goat polyclonal | R and D Systems | Cat# AF808; RRID:AB_2194992 | dilution 1:300 |
| Antibody | anti-calretinin (CALB2), rabbit polyclonal | Millipore | Cat# AB5054; RRID:AB_2068506 | dilution 1:500 |
| Antibody | anti-SMI32 (NEFH), mouse monoclonal | Covance | Cat# SMI-32R-100; RRID:AB_509997 | dilution 1:1000 |
| Antibody | anti-tyrosine hydroxylase (TH), rabbit polyclonal | Millipore | Cat# AB152; RRID:AB_390204 | dilution 1:1000 |
| Antibody | anti-Chat, goat polyclonal | MilliporeSigma | Cat# AB144P; RRID:AB_2079751 | dilution 1:500 |

*Continued on next page*

*Continued*

| Reagent type (species) or resource | Designation | Source or reference | Identifiers | Additional information |
|---|---|---|---|---|
| Antibody | anti-Aldoc, rabbit polyclonal | Dr. Izumi Sugihara; PMID:15470143 | RRID:AB_2313920 | dilution 1:8000 of 14.66 mg/mL stock |
| Antibody | anti-PLCb4, rabbit polyclonal | Santa Cruz Biotechnology | Cat# sc-20760 | dilution 1:100 |
| Antibody | anti-orexin, rabbit polyclonal | Peninsula Laboratories | Cat# T4074; RRID:AB_2315020 | dilution 1:500 |
| Antibody | anti-GFP, rabbit polyclonal | Millipore | Cat# AB3080; RRID:AB_91337 | dilution 1:1000 |
| Antibody | anti-GFP, chicken polyclonal | Aves Labs | Cat# GFP-1020; RRID:AB_2307313 | dilution 1:1000 |
| Antibody | anti-RFP, rabbit polyclonal | MBL International | Cat# PM005; RRID:AB_591279 | dilution 1:1000 |
| Recombinant DNA reagent | AAV1.hSyn.Cre.WPRE.hGH | Penn Vector Core | Lot. CS1087 | titer $4.37 \times 10^{13}$ GC/ml |
| Recombinant DNA reagent | AAV1.hSyn.eGFP.WPRE.bGH | Penn Vector Core | Lot.CS0500-3CS | titer $2.24 \times 10^{13}$ GC/ml |
| Recombinant DNA reagent | AAV1.hSyn.TurboRFP.WPRE.rBG | Penn Vector Core | Lot. V3731TI-R | titer $2.15 \times 10^{13}$ GC/ml |
| Recombinant DNA reagent | AAV2retro.CAG.Cre | UNC Vector Core | Lot. AV7703C | titer $5.3 \times 10^{12}$ GC/ml |
| Recombinant DNA reagent | AAV9.CAG.Flex.eGFP.WPRE.bGH | Penn Vector Core | Lot. CS0374 | titer $2.28 \times 10^{13}$ GC/ml |
| Recombinant DNA reagent | AAV9.CAG.Flex.tdTomato.WPRE.bGH | Penn Vector Core | Lot. CS0634 | titer $1.25 \times 10^{13}$ GC/ml |
| Recombinant DNA reagent | AAV9.CAG.hChR2(H134R)-mCherry.WPRE.SV40 | Penn Vector Core | Lot. CS0753-3CS | titer $1.71 \times 10^{13}$ GC/ml |
| Recombinant DNA reagent | AAV9.hSyn.eGFP.WPRE.bGH | Penn Vector Core | Lot. CS0354 | titer $8.88 \times 10^{13}$ GC/ml |
| Recombinant DNA reagent | AAV9.hSyn.TurboRFP.WPRE.rBG | Penn Vector Core | Lot. V4861MI-R | titer $6.64 \times 10^{13}$ GC/ml |
| Recombinant DNA reagent | RVdG-mC; glycoprotein deleted recombinant rabies virus carrying mCherry gene | GT3 core at the Salk Institute | | titer $9.6 \times 10^{9}$ TU/ml |
| Chemical compound, drug | dextran (3000 MW), conjugated with biotin | Thermo Fisher Scientific | Cat# D7135 | |
| Chemical compound, drug | dextran (3000 MW), conjugated with Texas Red | Thermo Fisher Scientific | Cat# D3328 | |
| Chemical compound, drug | red retrobeads IX | Lumafluor | | |
| Chemical compound, drug | green retrobeads IX | Lumafluor | | |
| Chemical compound, drug | FastBlue | Polysciences, Inc | Cat# 17740–1 | |

## Mice

C57BL/6J and YFP-16 transgenic line (*Feng et al., 2000*) backcrossed to C57BL/6J were used in single-cell harvesting at P15-34. C57BL/6J, YFP-16, GlyT2-EGFP (*Zeilhofer et al., 2005*), Gad2-nls-mCherry (stock 023140, Jackson Laboratory, Bar Harbor, ME), L7Cre (also known as Pcp2Cre, stock

004146, JAX), Ai3 (stock 007903, Jackson Laboratory), Ai14 (stock 007908, Jackson Laboratory), Ai32 (stock 024109, Jackson Laboratory), Sun1-sfGFP (stock 021039, Jackson Laboratory), VgluT2-Cre (stock 016963, Jackson Laboratory), Gad2Cre (stock 019022, Jackson Laboratory), ChatCre (stock 018957, Jackson Laboratory), and Lhx6-EGFP (stock 000246, MMRRC) mice were used in histological experiments (at >P50). Key resources table provides detailed information for the mice used. If needed, Cre-reporter mice were crossed with Ai3, Ai14, or Sun1-sfGFP reporter mice in which Cre recombination allows CAG promoter driven expression of EYFP, tdTomato, or nuclear membrane protein SUN1 fused superfolder GFP, respectively. All the experiments were performed under the protocols approved by Salk Institute Animal Care and Use Committee (11–00024) and Johns Hopkins University Animal Care and Use Committee (M014M28 and M016M464).

## Preparation of single-cell cDNA library

Single-cell qPCR and data analyses were performed as described previously (*Kodama et al., 2012*) with modifications in trehalose treatment, enzymes for tissue digestion, and calcein-AM cell staining. All chemicals were obtained from Sigma-Aldrich (St. Louis, MO) unless otherwise noted. Mouse brains were dissected out after decapitation under deep anesthesia with sodium pentobarbital (100 mg/kg). Acute cerebellar coronal slices were cut at 250 µm in ice-cold low Ca artificial cerebrospinal fluid (ACSF; 125 mM NaCl, 1.25 mM KCl, 25 mM NaHCO$_3$, 1.25 mM NaH$_2$PO$_4$, 1 mM CaCl$_2$, 3 mM MgCl$_2$, and 25 mM dextrose) with bubbling 95% O$_2$ and 5% CO$_2$ using a vibratome (VT1000 S, Leica Microsystems, Buffalo Grove, IL). Acute bilateral fastigial nuclei were quickly excised with a sharp knife from 2 or three cerebellar slices and were then enzymatically digested with cysteine (2 mM)-supplemented papain (40 U/ml, Worthington, Lakewood, NJ) and chondroitinase-ABC (1 U/ml) in the 37°C incubation solution, which was low Ca HEPES-buffered ACSF (140 mM NaCl, 1.25 mM KCl, 10 mM HEPES, 1.25 mM NaH$_2$PO$_4$, 1 mM CaCl$_2$, 3 mM MgCl$_2$, and 25 mM dextrose) containing blockades (0.1 µM tetrodotoxin, 20 µM 6-cyano-7-mitroquinozaline-2,3-dione, and 50 µM D-APV) and trehalose (5%). Digestion was quenched by transferring the tissues into the ice-cold incubation solution containing bovine serum albumin (1%). The tissues were triturated by gentle pipetting and the suspension was then transferred in a dish with ice-cold incubation solution. To visualize cells from wild type mice, cells were fluorescently stained by adding calcein-AM in the digestion tube (40 µM), quencher tube (40 µM), and the first dish (8 µM). Single cells were manually picked up with pulled glass capillaries under a fluorescent stereoscope (M205 FA, Leica Microsystems) attached with a high-power light source (PhotoFluor II NIR, 89 NORTH, Williston, VT). They were immediately transferred to the second dish containing the same solution to wash any residual debris. From the second dish, single cells were transferred using pulled glass capillaries to individual tubes containing cell lysis buffer and spike-in RNAs (Lys, Trp, Phe, and Thr; 1000, 100, 20, and five copies, respectively). As negative controls, we put the following materials instead of cells; 10 pg of mouse brain total RNA as positive control, and the incubation solution from the second dish after the harvesting as negative control. The cDNAs were globally amplified by performing cell lysis, reverse transcription, exonuclease I treatment, poly(dA) addition, second-strand cDNA synthesis, and 20 cycle PCR reaction as described in *Kurimoto et al., 2007*; *Kodama et al., 2012*. Note that this method amplifies ~700 bp of the 3' end of the cDNAs.

## qPCR, sample curation, and gene expression profiling

Primers/probes for qPCR were designed for 3'-end sequences avoiding polyadenylation signals as previously described (*Kodama et al., 2012*) and purchased from Integrated DNA Technologies, Inc (IDT, Coralville, IA). The PrimerQuest Tool available at IDT's website was also used for designing primers. Their sequences are available in *Figure 2—source data 1*.

All samples were initially screened for *Gapdh*, a housekeeping gene, for quality control. A 10 µL reaction scale TaqMan method qPCR for Gapdh was performed using 1 µL of 1:40 diluted 20 cycle PCR products with 7900HT Fast Real-Time PCR System (Applied Biosystems, Foster City, CA) or CFX384 Touch Real-Time PCR Detection System (Biorad, Hercules CA). When *Gapdh* expression was detected in the negative control, the experiment batch was regarded of as contaminated and the samples were excluded from the study (2 batches out of 16, which included 34 cells in total). Samples that showed unexpectedly high *Gapdh* Ct values (>mean + SD), suggesting degradation of mRNAs, were also excluded. The single cell cDNA samples that passed the quality control with

*Gapdh* were further amplified by 11 cycle PCR, as described in *Kodama et al., 2012*, to obtain large cDNA libraries for the following gene expression profiling with qPCR. Linear amplification of spike-in genes in extra-amplified samples inferred similarly linear amplification of other genes in the cDNA libraries. The samples of glutamatergic neurons were curated by performing qPCR to examine the absence of glial genes, *Mobp* and *Cd68*, to examine the presence of neuronal genes, *Scn8a* and *Kcna1*, to examine the absence of inhibitory neuronal genes, *Gad1*, *Gad2*, and *Slc6a5*, and then to examine the presence of glutamatergic neuronal gene, *Slc17a6*. The qPCR for gene expression profiling was performed in triplicated 10 µL scale reactions. Each reaction plate had at least four wells of standard reaction which was cDNA of 1 pg Trp. The threshold to determine Ct was set so that Ct values of the standard reaction became identical across all the reaction plates. Averaged Ct values, without normalization, were used for the following analysis.

## qPCR data analysis

The analysis of the expression data was performed in R (https://www.r-project.org/) and Igor (Wave-Metrics, Portland OR) software. The data were subjected to the hierarchical clustering analysis without normalization. To eliminate influence of fluctuating stochastic gene expression at low levels, the Ct values greater than the average of spike-in RNA Thr (23.43; corresponding to five copies input mRNA) were regarded as not determined (ND). For clustering, euclidean distance was calculated between any pairs of samples and hierarchical clustering was performed with Ward's method using 'ward.D2' method in 'hclust()' function in R.

## Tracer injection and histology

Stereotaxic tracer injections were made into adult mice. AAV-mediated anterograde transsynaptic tracing was performed in the Ai14 mouse line (*Zingg et al., 2017*). For anterograde axonal tracing, we used AAV9.hSyn.TurboRFP.WPRE.rBG (titer $6.64 \times 10^{13}$ GC/ml), AAV9.hSyn.eGFP.WPRE.bGH (titer $8.88 \times 10^{13}$ GC/ml), AAV1.hSyn.eGFP.WPRE.bGH (titer $2.24 \times 10^{13}$ GC/ml), AAV1.hSyn.TurboRFP.WPRE.rBG (titer $2.15 \times 10^{13}$ GC/ml), AAV9.CAG.Flex.eGFP.WPRE.bGH (titer $2.28 \times 10^{13}$ GC/ml), AAV9.CAG.Flex.tdTomato.WPRE.bGH (titer $1.25 \times 10^{13}$ GC/ml), and AAV9.CAG.hChR2 (H134R)-mCherry.WPRE.SV40 (titer $1.71 \times 10^{13}$ GC/ml) (all from Penn Vector Core at University of Pennsylvania). For anterograde transsynaptic tracing, we used AAV1.hSyn.Cre.WPRE.hGH (titer $4.37 \times 10^{13}$ GC/ml) (from Penn Vector Core). For retrograde tracing, we used FastBlue (5% in distilled water, 17740–1, Polysciences, Inc, Warrington, PA), fluorescent latex microspheres (red retrobeads IX or green retrobeads IX, Lumafluor, Naples FL), AAV2retro.CAG.Cre (titer $5.3 \times 10^{12}$ GC/ml, UNC Vector Core at University of North Carolina, Chapel Hill, NC) dextrans conjugated with Texas Red (10% in saline, D3328, Thermo Fisher Scientific) and with biotin (BDA, MW3000, 10% in saline, D7135, Thermo Fisher Scientific), and glycoprotein deleted recombinant rabies virus carrying mCherry gene (RVdG-mC) (titer $9.6 \times 10^9$ TU/ml, from GT3 core at the Salk Institute). A mouse was placed on the stereotaxic apparatus (David Kopf Instruments, Tujunga, CA) under isoflurane anesthesia (2% in 1.0 L/min oxygen). A small incision was made in the head skin overlying the skull of the mouse and a small craniotomy was drilled in the skull with a dental drill. Pulled glass pipette (tip inner diameter of 5–15 µm) filled with a tracer was stereotaxically inserted in the brain using Angle One stereotaxic instrument (myNeuroLab.com; currently Leica Instruments) and a small amount of the tracer (5–300 nL) was pressure injected using Picospritzer II (Parker Hannifin, Hollis, NH) with nitrogen (15–30 psi) at the target. For the tracer injections into the FN, local field potential was also monitored through extracellular amplifier (ER-1, Cygnus, bandpass frequency range 1–3000 Hz, Cygnus Technology, Delaware Water Gap, PA), 50/60 Hz noise eliminator (HumBug, Quest Scientific, North Vancouver, BC), and a speaker. Injections to the cerebellar cortex were made in specific lobules which were identified through a craniotomy. Ten minutes after the injection, the glass pipette was removed from the brain and the skin was sutured. The mice survival times were 3 weeks for AAV axonal tracing, 4 weeks for AAV transsynaptic somatic labeling, 5 weeks for AAV transsynaptic axonal tracing, 4–7 days for retrobead, dextran, and FastBlue, and 4–5 days for RVdG-mC.

After appropriate survival periods, mice were deeply anesthetized with 2,2,2-tribromoethanol (also known as avertin, 0.5 mg/g) and were then transcardially perfused with PBS containing heparin (10 U/ml) followed by 4% paraformaldehyde (PFA) in PBS. Brains were dissected out from the skull, postfixed overnight in 4% PFA in PBS and cryoprotected in 30% sucrose PBS. Those brains were

then embedded in gelatin in order to prevent separation of the cerebellum from the brainstem. The gelatin blocks were hardened on ice and were then trimmed and fixed overnight with 4% PFA in 30% sucrose PFA. Coronal or sagittal serial sections were cut at a thickness of 40 µm with freezing microtome. After rinsed with PBS containing 0.15% Triton-X (PBST), the sections were stored in PBS containing 0.1% sodium azide at 4°C until use.

For immunostaining, sections were incubated overnight on a shaker at 4°C with primary antibodies and 5% normal donkey serum (NDS) in PBST. Primary antibodies used were mouse anti-alpha synuclein (1:100, 610786, BD Bioscience, San Jose, CA), goat anti-osteopontin (1:300, AF808, R and D Systems, Minneapolis, MN), rabbit anti-calretinin (1:500, AB5054, Millipore), rabbit anti-tyrosine hydroxylase (1:1000, AB152, Millipore, Burlington, MA), rabbit anti-GFP (1:1000, AB3080, Millipore), chicken anti-GFP (1:1000, GFP-1020, Aves Labs), rabbit anti-RFP (1:1000, PM005, MBL International, Woburn, MA), goat anti-Chat (1:500, AB144P, MilliporeSigma), rabbit anti-PLCb4 (1:100, sc-20760, Santa Cruz Biotechnology, Dallas TX), rabbit anti-Aldoc (1:8000 of 14.66 mg/mL stock, Dr. Izumi Sugihara), rabbit anti-orexin (1:500, T4074, Peninsula Laboratories, San Carlos, CA), mouse anti-SMI32 (NEFH, 1:1000, SMI-32R-100, Covance, Princeton, NJ). After being rinsed five times with PBST for more than an hour in total, sections were incubated overnight at 4°C with donkey secondary antibodies that had been processed for minimum cross reactivity by the manufacturer and were conjugated with AlexaFluor488, AlexaFluor594, or AlexaFluor647 fluorescent dye (Jackson Immunoresearch) and 5% NDS in PBST. BDA was visualized with streptavidin conjugated with AlexaFluor488 or AlexaFluor594 dye. Some sections were counterstained by adding fluorescent Nissl (NeuroTrace 435/455 or 640/660, Thermo Fisher Scientific, Waltham, MA) or DAPI to the incubation with secondary antibody. The stained sections were then rinsed five times with PBS for more than an hour in total, mounted on slide glass, and coverslipped with VECTASHIELD (H-1400, Vector Laboartories, Burlingame, CA) or Fluoroshield (F6937, MilliporeSigma).

Notably, we found that the anterograde transsynaptic tracing method by *Zingg et al., 2017*, could also robustly visualize the axons of postsynaptic (second order) neurons with a long (5 weeks) post-injection survival period. Thus, we applied this modified method to map disynaptic input from the FN to the forebrain (*Figure 6*). Although the transneuronal viral transport is reported to be only be monosynaptic, our injections occasionally labeled presumed third order neurons (e.g. cortical pyramidal neurons). This might be because we injected 10-fold more AAV copies than in the original report (*Zingg et al., 2017*). The effect was, however, estimated to be very minor (for example, only a few labeled pyramidal neurons were found across the entire cortex per case, see *Figure 6—figure supplement 1*) and axonal labeling from those tertiary neurons was regarded to have little or no impact on the interpretation of disynaptic projections.

## Anatomical data analysis

We used the Paxinos mouse brain atlas (*Paxinos and Franklin, 2012*) for anatomical nomenclature, supplemented by nomenclature based on the literature. Images were taken with a confocal microscope system (FV1000, Olympus, Tokyo, Japan) or a CCD camera (ORCA-100, Hamamatsu Photonics, Hamamatsu, Japan) attached to an epi-fluorescent microscope (BX61, Olympus). Contrast and brightness were adjusted in Photoshop CS6 (Adobe, San Jose, CA). In some panels, pseudo-low magnification images were made by tiling several photos taken at higher magnification.

To analyze whether glutamatergic, glycinergic, and GABAergic fastigial neurons express SPP1, SNCA, and CALB2, we performed immunostaining for SPP1, SNCA, and CALB2 of fastigial sections of the reporter mouse lines of glutamatergic, glycinergic, and GABAergic neurons, which are VgluT2-Cre;Ai14, GlyT2-EGFP, and Gad2-nls-mCherry. Images of the fastigial subregions were taken with single confocal scans using a 40x objective lens. Cell-counting was performed only when the nucleus of the neurons was identified.

Neuronal size measurements were analyzed from epifluorescent microphotographs of individual neurons taken with a 20x objective lens with the focus centered at the neuronal nucleus. Neuronal somata were manually outlined with the measurement tool in ImageJ (https://imagej.nih.gov/ij/), which calculates the area of each circumscribed region.

Axon diameters were assessed with ImageJ from confocal micrographs of AAV-injected FN. Confocal z-stack images of the entire thickness of each section (40 µm) were taken with a 100x lens at the level of the superior cerebellar peduncle. To reliably image axons, which typically are >0.75 µm, we used z-stack steps of 0.36 µm. Axon diameter was calculated as the average of 3 points

measured from each axon. Measurements were obtained only from smooth axons that did not exhibit degenerative morphology such as swelling.

For each tracing experiment, injection site was carefully checked for the spread of the tracers to neighboring (sub)nuclei. Only the cases without those spread, which were listed in *Figure 3—source data 2*, were included in the analyses.

Mapping of retrogradely traced fastigial neurons was performed on our standard map of the fastigial nucleus which was obtained from Nissl stained serial sections of an adult mouse. One in three 40 µm thick sections from each injection case was used for the mapping. Correspondence of rostro-caudal levels between the standard map and the sample section was confirmed by assigning relative coordinates of 0% being caudal end and 100% being rostral end of the FN. Each of the retrogradely labeled neurons was mapped. For figure panels, three representative levels were chosen from the standard map and the mapping results were combined on the drawings of corresponding levels.

For analysis of PCs stripes in cerebellar zones, an anatomical atlas produced in previous studies was used (*Sugihara and Shinoda, 2004*; *Fujita et al., 2014*). The scheme of Aldoc expression map on the unfolded cerebellar cortex was redrawn based on *Fujita et al., 2014*. The pattern of PC stripes is highly reproducible across individuals and across mammalian species (*Hawkes and Leclerc, 1987*; *Sillitoe et al., 2005*; *Apps and Hawkes, 2009*; *Fujita et al., 2010*; *Marzban and Hawkes, 2011*; *Fujita et al., 2014*), and each of the stripes has its established nomenclature (e.g. 1+, 2+, a+, etc.). Since individual IO neurons typically project to two distinct stripes that are separately located and termed in the anterior and posterior lobes of the cerebellum (*Fujita and Sugihara, 2013*), these pairs of stripes are referred to as, for example, a+//2+, in which a+ and 2+ indicate the stripes in anterior and posterior lobes, respectively, and '//' indicates anteroposterior pairing of the stripes. Identification of the distinct Purkinje cell (PC) populations was based on the mediolateral level from the midline and the immunoreactivity for marker molecules, Aldoc or PLCb4 (*Sarna et al., 2006*; *Fujita et al., 2014*). Subdivisions of inferior olivary subnuclei were identified by referring to *Azizi and Woodward, 1987*; *Yu et al., 2014*. In addition, we designated an additional subarea, which we termed subnucleus 'd' of the caudal medial accessory olive, based on input connectivity with the superior colliculus (*Akaike, 1992*; *Figure 7—figure supplement 2*) and cholinergic axons (*Yu et al., 2014*) as well as output connectivity with distinct Purkinje cells that were located at lobule VIc/VII, Crus I or Crus II (*Akaike, 1986*; *Apps, 1990*; *Figure 7—figure supplement 2*).

For mapping the disynaptic input to the cortex, we used the borders of cortical functional areas in *Watson et al., 2012* with some simplification. The density of labeled axons at layer one or at deeper layers in each functional area was manually and qualitatively assessed and then those areas were categorized in three schemes as dense, moderate, or not significant. Color-coding was performed based on these three categories.

## Acknowledgements

We thank Shiloh Guerrero, Burhan Jama, and Minh Lam for assistance with qPCR experiments, and Matthew Ehrenburg for assistance with histological analyses. Antibody for Aldoc and sections of Aldoc-venus mouse were generous gifts from Dr. Izumi Sugihara. This research was supported by NIH grants NS095232 and NS105039 to SdL, and JSPS Overseas Research Fellowship (26-585) to HF.

## Additional information

### Funding

| Funder | Grant reference number | Author |
| --- | --- | --- |
| National Institute of Neurological Disorders and Stroke | NS095232 | Sascha du Lac |
| National Institute of Neurological Disorders and Stroke | NS105039 | Sascha du Lac |
| Japan Society for the Promotion of Science | 26-585 | Hirofumi Fujita |

The funders had no role in study design, data collection and interpretation, or the decision to submit the work for publication.

## Author contributions
Hirofumi Fujita, Conceptualization, Data curation, Formal analysis, Funding acquisition, Validation, Investigation, Visualization, Methodology, Writing - original draft, Writing - review and editing; Takashi Kodama, Methodology, Writing - review and editing; Sascha du Lac, Conceptualization, Data curation, Formal analysis, Supervision, Funding acquisition, Validation, Investigation, Methodology, Writing - original draft, Writing - review and editing

## Author ORCIDs
Hirofumi Fujita (ID) https://orcid.org/0000-0001-9630-2756
Takashi Kodama (ID) http://orcid.org/0000-0003-3110-979X
Sascha du Lac (ID) https://orcid.org/0000-0003-4669-3191

## Ethics
Animal experimentation: All procedures conformed to NIH guidelines and were approved by the Johns Hopkins University Animal Care and Use Committee (M014M28 and M016M464) and Salk Institute Animal Care and Use Committee (11-00024).

## Decision letter and Author response
Decision letter https://doi.org/10.7554/eLife.58613.sa1
Author response https://doi.org/10.7554/eLife.58613.sa2

# Additional files
## Supplementary files
• Transparent reporting form

## Data availability
All data analyzed during this study are included in the manuscript and supporting files.

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
