## [Decision Letter]

**Acceptance summary:**

The reviewers found this to be a very impressive study that identifies different cell types within the fastigial nucleus (medial cerebellar nucleus) and maps the fastigial heterogeneity to cerebellar modules and connectivity to extra-cerebellar targets. The work is impressive in detail and amount, and its implications are of great importance to a large audience working in the cerebellar field. The revised manuscript very nicely addressed all of the issues raised with the initial submission and reviewers congratulate the authors on a wonderful paper.

**Decision letter after peer review:**

Thank you for submitting your manuscript "Modular output circuits of the fastigial nucleus mediate diverse motor and nonmotor functions of the cerebellar vermis" for consideration by *eLife*. Three peer reviewers reviewed your manuscript, one of whom is a member of our Board of Reviewing Editors, with their evaluation overseen by Richard Ivry as the Senior Editor. The reviewers and the Reviewing Editor drafted this decision letter to help you prepare a revised submission.

Summary:

Reviewers found this to be a very impressive study that identifies different cell types within the fastigial nucleus (medial cerebellar nucleus) and maps the fastigial heterogeneity to cerebellar modules and connectivity to extra-cerebellar targets. The manuscript addresses a seminal question, laying a foundation for performing studies on the other cerebellar nuclei and for future functional studies on how different modules influence specific neural networks and behaviors. The work is impressive in detail and amount, and its implications are of great importance to a large audience working in the cerebellar field.

Essential Revisions:

1) Title and Abstract and should be toned down. While there are functional implications of the study, no motor or non-motor functions/behaviors were actually tested. See 5 below for more detail.

2) Provide additional experimental details:

A) Indicate the number and sex of mice used for RNA-seq and synaptic connection clearly in figure legends. Where any variations noted based on male versus female?

B) Provide a list of all anatomical coordinates and injection volumes that they used for their tracing experiments (both anterograde and retrograde). The results suggest areas can precisely targeted as small as 0.5 mm in diameter (which is very impressive). The Materials and methods describe that injections were 5-300 nL, which makes it important to be specific since such orders of magnitude in different volumes could be critical at this scale. Also, 300 nL is significantly larger than a 0.5*0.5*0.5 mm volume. Please discuss.

C) Give a more detailed description of the reproducibility of the findings. Short of Figure 4A, injection sites are not provided. Somewhat related to above, how comparable was this across animals? It would be helpful to see how reproducible the injections were across different mice.

D) Given the variability of viral infections and stereotaxic injections, it is expected that there would be some variability in results across biological replicates. Are the target areas of projections defined as only those areas that receive a predominant set of projections in all replicates? In other words, how was animal to animal variability accounted for, and what is the baseline projection that is considered representative? This needs to be stated in the text and figure legends.

3) Data presentation:

Most images for the local anterograde tracing experiments are heavily cropped (again, Figure 4A is a beautiful exception) and while it is claimed that widespread anterograde projections to each sub-region, only a handful are presented in each figure. While not all the data can be easily shown, would be very useful to show the full range of terminal fields that resulted from the tracing.

4) Definition of the five subgroups:

Separation of the rDLP and rFN into two regions appears arbitrary. In Figure 1 has an arbitrary anatomical boundary that is not entirely consistent with the marker expression. In Figure 2, there is no separation between these groups. In Figure 4, there is a high degree of overlap in projection patterns from the rDLP and rFN region. And in Figure 7, they show that the rDLP and rFN both receive input from Aldoc-negative PCs, albeit from different medio-lateral zones (which is to be expected given that rDLP and rFN are situated in different medio-lateral regions). If the aim is to segregate these two regions, it would be beneficial to make a stronger argument for doing so. Also, some justification must be provided for the boundaries chosen. Also, it may be helpful to present Figure 7 earlier in the manuscript (at least before Figure 4), as Figure 7 provides the strongest evidence that rDLP and rFN are integrated in different anatomical modules.

It would also be helpful to be consistent in calling anatomical regions by one name throughout the manuscript; either rFN or F1R but not both and not interchangeably (it is hard to follow for someone less familiar with this subject). If there are differences in these two nomenclatures, it would be helpful to carefully describe the differences. The color coding is appreciated in the figures, but consistency in nomenclature would be beneficial in the text too.

5) Interpretation:

While are thorough job of defining the anatomical connectivity patterns was given there are no experimental tests that link these anatomical findings to computational, electrophysiological, and behavioral outputs. Thus, all claims about function are purely speculative, beginning with a highly suggestive title, and ending with a highly speculative discussion. I would suggest that the authors focus their discussion on the anatomical and structural connectivity findings rather than make broad suggestions that are not tested in this study. The anatomical findings are very interesting by themselves and the speculation is therefore a distraction.

---

## [Author Response]

Essential Revisions:1) Title and Abstract and should be toned down. While there are functional implications of the study, no motor or non-motor functions/behaviors were actually tested. See 5 below for more detail.

We toned down the title and the last sentence of the Abstract.

2) Provide additional experimental details:A) Indicate the number and sex of mice used for RNA-seq and synaptic connection clearly in figure legends. Where any variations noted based on male versus female?

We now include the number and sex of mice in the relevant figure legends. For the qPCR gene expression analyses, only males were used. However, we expect that the expression of the genes used for our qPCR analyses are similar between males and females, because 1) for the marker genes, we did not find any sex differences in the immunostainings, which were performed in combination with retrograde tracing, and 2) for the genes related to neuronal firing, these gene expressions in the vestibular neurons, which share several features with the FN (as described in the subsection “Single cell gene expression confirms anatomically distinct cell types”), are known to be similar between male and female mice (Kodama et al., 2020).

For the tracing experiments, although this study was not designed to carefully assess sex differences, we did not find any qualitative variations between male and female neuronal connections. Sexes used in each experiment are now indicated in figure legends and listed in Figure 3—figure supplement 4.

B) Provide a list of all anatomical coordinates and injection volumes that they used for their tracing experiments (both anterograde and retrograde). The results suggest areas can precisely targeted as small as 0.5 mm in diameter (which is very impressive). The Materials and methods describe that injections were 5-300 nL, which makes it important to be specific since such orders of magnitude in different volumes could be critical at this scale. Also, 300 nL is significantly larger than a 0.5*0.5*0.5 mm volume. Please discuss.

We now provide a summary table for the experimental parameters for injection experiments in Figure 3—figure supplement 4. The wide range of injection volume is due to the difference in the size of the target and the tracers used. The upper limit of the injection volume is determined by the size of the target and the presence of fastigial axons or terminals that surround the target.

We also have written a Discussion section "Technical consideration" to discuss some limitations of the tracing techniques used in this study.

C) Give a more detailed description of the reproducibility of the findings. Short of Figure 4A, injection sites are not provided. Somewhat related to above, how comparable was this across animals? It would be helpful to see how reproducible the injections were across different mice.D) Given the variability of viral infections and stereotaxic injections, it is expected that there would be some variability in results across biological replicates. Are the target areas of projections defined as only those areas that receive a predominant set of projections in all replicates? In other words, how was animal to animal variability accounted for, and what is the baseline projection that is considered representative? This needs to be stated in the text and figure legends.

In response to C) and D): Thank you for these important points. We did not observe a high degree of variability in the results of the circuit tracing experiments.

Sources of variation in the tracing results in this study result derive from 1) variations in injection centers and tracer spread, 2) inter-individual differences in anatomical circuits, and 3) variable tracer efficiency for certain cell types such as virus tropism. For 1), we excluded the injection cases whose tracer spread were not confined within the targeted (sub)nuclei. For 2), fastigial output circuits were remarkably consistent across individuals, as n=6 pan-injection cases for anterograde fastigial axonal labeling resulted in the same results as now described in the subsection “Downstream targets of excitatory fastigial cell types are distinct”, and shown in Figure 3—figure supplement 2. For 3), we confirmed pan and localized anterograde tracing with both virus (AAV) and dye (BDA) tracers which both resulted in the same results. These points are now discussed in the first paragraph of a new Discussion section "Technical considerations".

An example of reproducibility of the results are now shown in new Figure 5—figure supplement 1, and figure legend of this figure now mention similarity in the two injection cases shown.

) Data presentation:Most images for the local anterograde tracing experiments are heavily cropped (again, Figure 4A is a beautiful exception) and while it is claimed that widespread anterograde projections to each sub-region, only a handful are presented in each figure. While not all the data can be easily shown, would be very useful to show the full range of terminal fields that resulted from the tracing.

We have added Figure 4—figure supplement 1 and Figure 5—figure supplement 1 to show representative axonal trajectory of the projections from fastigial subregions.

4) Definition of the five subgroups:Separation of the rDLP and rFN into two regions appears arbitrary. In Figure 1 has an arbitrary anatomical boundary that is not entirely consistent with the marker expression. In Figure 2, there is no separation between these groups. In Figure 4, there is a high degree of overlap in projection patterns from the rDLP and rFN region. And in Figure 7, they show that the rDLP and rFN both receive input from Aldoc-negative PCs, albeit from different medio-lateral zones (which is to be expected given that rDLP and rFN are situated in different medio-lateral regions). If the aim is to segregate these two regions, it would be beneficial to make a stronger argument for doing so. Also, some justification must be provided for the boundaries chosen. Also, it may be helpful to present Figure 7 earlier in the manuscript (at least before Figure 4), as Figure 7 provides the strongest evidence that rDLP and rFN are integrated in different anatomical modules.

The distinction between rDLP and rFN is supported by previous research. To clarify this in the manuscript, we now start the Results with following introductory paragraph: "The fastigial nucleus (FN) is the medial cluster of cerebellar nucleus neurons (Dow, 1942), delineated in mammals by surrounding white matter (Larsell, 1970). […] Each of these subdivisions have been linked with distinct sets of Purkinje cells, inferior olive subnuclei, and downstream brain regions (Batton et al., 1977; Teune et al., 2000; Voogd and Ruigrok, 2004; Sugihara and Shinoda, 2007; Sugihara et al., 2009)."

It would also be helpful to be consistent in calling anatomical regions by one name throughout the manuscript; either rFN or F1R but not both and not interchangeably (it is hard to follow for someone less familiar with this subject). If there are differences in these two nomenclatures, it would be helpful to carefully describe the differences. The color coding is appreciated in the figures, but consistency in nomenclature would be beneficial in the text too.

To be consistent with terminology for fastigial subregions, we have revised the entire text to use rFN, rDLP, cDLP, vlFN, and cFN.

5) Interpretation:While are thorough job of defining the anatomical connectivity patterns was given there are no experimental tests that link these anatomical findings to computational, electrophysiological, and behavioral outputs. Thus, all claims about function are purely speculative, beginning with a highly suggestive title, and ending with a highly speculative Discussion. I would suggest that the authors focus their Discussion on the anatomical and structural connectivity findings rather than make broad suggestions that are not tested in this study. The anatomical findings are very interesting by themselves and the speculation is therefore a distraction.

We have modified the title and statements in the Abstract, Discussion, and figure legends which overstated the links between our structural findings and their functional implications. The discussion of neuronal connectivity in each module summarizes a complex and large amount of literature which has not been previously contextualized. We believe that the hypothetical functional framework provided in Figure 8 and the Discussion will have significant heuristic value for the field.